

# The Flexible Ocean and Climate Infrastructure Version 1 (FOCI1): Mean State and Variability

Katja Matthes[1,2], Arne Biastoch[1,2], Sebastian Wahl[1], Jan Harlaß[1], Torge Martin[1], Tim Brücher[1], Annika Drews[1], Dana Ehlert[1], Klaus Getzlaff[1], Fritz Krüger[1], Willi Rath[1], Markus Scheinert[1], Franziska U. Schwarzkopf[1], Tobias Bayr[1], Hauke Schmidt[3], and Wonsun Park[1]

[1]GEOMAR Helmholtz Centre for Ocean Research Kiel, Kiel, Germany
[2]Christian-Albrechts Universität zu Kiel, Kiel, Germany
[3]Max-Planck Institute for Meteorology, Hamburg, Germany

*Correspondence to:* K. Matthes (kmatthes@geomar.de)

**Abstract.** A new Earth system model, the Flexible Ocean and Climate Infrastructure (FOCI), is introduced. A first version of FOCI consists of a global high-top atmosphere (ECHAM6.3) and an ocean model (NEMO3.6) as well as sea ice (LIM2) and land surface model components (JSBACH), which are coupled through the OASIS3-MCT software package. FOCI includes a number of optional modules which can be activated depending on the scientific question of interest. In the atmosphere, inter-
active stratospheric chemistry can be used (ECHAM6-HAMMOZ) to study, for example, the effects of the ozone hole on the climate system. In the ocean, a biogeochemistry model (MOPS) is available to study the global carbon cycle. A unique feature of FOCI is the ability to explicitly resolve mesoscale ocean eddies in specific regions. This is realized in the ocean through nesting; first examples for the Agulhas Current and the Gulf Stream systems are described here. FOCI therefore bridges the gap between coarse-resolution climate models and global high-resolution weather prediction and ocean-only models. It allows
to study the evolution of the climate system on regional and seasonal to (multi-) decadal scales.

The development of FOCI resulted from a combination of the long-standing expertise in ocean and climate modeling in several research units and divisions at GEOMAR. FOCI will thus be used to complement and interpret long-term observations in the Atlantic, enhance the process understanding of the role of mesoscale oceanic eddies for large-scale oceanic and atmospheric circulation patterns, study feedback mechanisms with stratospheric processes, estimate future ocean acidification, improve the
simulation of the Atlantic Meridional Overturning Circulation changes and their influence on climate, ocean chemistry and biology.

In this paper we present both the scientific vision for the development of FOCI as well as some technical details. This includes a first validation of the different model components using several configurations of FOCI. Results show that the model in its basic configuration runs stably under pre-industrial control as well as under historical forcing, and produces a mean climate and
variability which compares well with observations, reanalysis products and other climate models. The nested configurations reduce some long-standing biases in climate models and are an important step forward to include the atmospheric response in multi-decadal eddy-rich configurations.



# 1 Introduction

In the light of international climate targets to limit global warming to 1.5° or 2°C, it is becoming increasingly important to provide more reliable information on the evolution of the climate system by representing its complexity on regional spatial and seasonal-to-decadal temporal scales. These scales are particularly important to improve the understanding of climate variabil-
ity and the adaptation of global warming – one of today's most pressing societal challenges. In particular information about consequences for European climate in the coming decades are needed from reliable climate model simulations which address the key drivers for regional changes.

Current climate models participating in Climate Model Intercomparison Projects (CMIPs) are designed for (multi-) centennial simulations and hence traditionally have a coarse, i.e. 1°-2°, resolution in the atmosphere and the ocean, restricting their relia-
bility in particular on regional scales. Due to computational limits some processes are neglected, such as the upper atmosphere (strato- and mesosphere), atmospheric chemistry as well as mesoscale eddies in the ocean that provide an important contri-bution not only to the ocean circulation but also to atmosphere-ocean interactions on global and particularly regional scales. Nevertheless, high-performance computing capacities as well as numerical methods for efficient dynamical codes of climate models increase and so does the possibility to run higher spatial resolution climate models globally (e.g., Delworth et al., 2012;
Bacmeister et al., 2014; Small and et al., 2014; Haarsma et al., 2016; Williams and et al., 2017; Müller et al., 2018). These recent high-resolution modeling studies have demonstrated the added value of increased resolution in particular for regional climate information (Haarsma et al., 2016). Watterson et al. (2014) compared the skill of Earth system models participating in phases 3 and 5 of the Climate Model Intercomparison Project (CMIP) and confirmed the link between better skill and finer horizontal resolution.

However, these very high resolution model simulations, resolving e.g. mesoscale ocean eddies, can still not be run for (multi-) centennial simulations or in series of integrations (ensembles). As part of the upcoming IPCC report, there is the High Res-olution Model Intercomparison Project (HighResMIP) for CMIP6 which presents for the first time a common protocol for high resolution runs with grid spacings of at least 50 km in the atmosphere and 0.25° in the ocean over the period 1950-2050 (Haarsma et al., 2016). This will provide a robust assessment of the benefits of increased horizontal resolution for climate sim-
ulations and address the question of how model biases are related to unresolved processes in the atmosphere and the ocean in a multi-model framework. Only very few of the participating modeling groups in HighResMIP will go to horizontal resolution in the ocean below 0.25°, where mesoscale ocean dynamics come into play. The importance of resolving western boundary currents, such as the Gulf Stream in the North Atlantic or the Kuroshio in the North Pacific, for a more realistic representation of ocean-atmosphere interactions and their effects on climate has been shown in a number of publications (e.g., Minobe et al.,
2008; Ma et al., 2016; Griffies et al., 2015; Renault et al., 2016; Omrani et al., 2019).

Clearly, depending on the scientific question of interest, a compromise is needed to bridge the gap between coarse climate models participating in CMIPs and global high-resolution numerical weather prediction models or eddy-resolving ocean-only models, respectively. Our genuine scientific interest lies in the role of the ocean in the climate system, in particular in deci-phering internal and external processes driving past, present and future ocean circulation and its role in climate on seasonal





and decadal to multi-decadal timescales. Specific research areas are 1) the fundamental understanding of the ocean system, 2) advanced prediction and attribution of changes in the ocean and in the climate system, and 3) understanding of the physical drivers of ecosystems and biogeochemical cycles. Therefore a model system that is able to resolve mesoscale ocean eddies, stratosphere-troposphere-ocean interactions, as well as ocean-biogeochemistry is required.

We aim to improve the understanding of the interaction of internal variability modes such as the El Niño Southern Oscillation (ENSO), the Pacific Decadal Oscillation (PDO), the Atlantic Meridional Overturning Circulation (AMOC), the Atlantic Multidecadal Variability (AMV), the North Atlantic Oscillation (NAO) and regional decadal climate variability and extremes. The attribution of recent extreme events such as the summer heat wave in 2018 and its (oceanic) drivers or the impact of enhanced melting of the Greenland Ice Sheet on ocean circulation and sea level rise will be an important focus in the future. Another

open question is the poor reproduction of coastal ocean upwelling systems in the Tropics and mid-latitudes, mainly located at the eastern ocean boundaries, in current Earth system models. These upwelling systems host the most productive ecosystems in the world and are thus fundamentally important for fisheries and food security. They also have been subject to large natural variability on time scales from weeks to decades and centuries, but the consequences of ongoing climate change are not well understood.

By combining the in-house knowledge and expertise, the modeling team at GEOMAR has spent a considerable effort in order to develop a new coupled model system which is based on the MPI-ESM, the NEMO community ocean model as well as other existing model components to fulfill the above requirements. The model system is characterized by a flexible structure and allows to resolve processes relevant for specific scientific question in one modeling framework. The overarching goal has been to combine, strengthen and facilitate inter-disciplinary (modeling) studies within GEOMAR and with external collaborators.

The result of this effort is the Flexible Ocean and Climate Infrastructure (FOCI) which is presented here.

Scientifically, FOCI builds upon our 10 year experience with the Kiel Climate Model (KCM, Park et al., 2009). Amongst other topics, the KCM has been used extensively to study internal and external climate variability from the past to future. This includes El Niño Southern Oscillation (ENSO, e.g., Bayr et al., 2018; Wengel et al., 2018; Latif et al., 2015), Atlantic mean state (biases) and their impact on variability (e.g., Harlaß et al., 2018, 2015; Wahl et al., 2009; Ding et al., 2015; Drews

et al., 2015) or Atlantic meridional overturning circulation variability (e.g., Park and Latif, 2008; Ba et al., 2013; Martin et al., 2015). Paleo-applications include time-slice and transient simulations for the Holocene and the Eemian (e.g., Schneider et al., 2010; Khon et al., 2018), the Last Glacial Maximum (e.g., Song et al., 2019), Pliocene (e.g., Krebs and Schneider, 2011; Song et al., 2017) and the Cretaceous (e.g., et al., 2019). A complete list of publications with the KCM can be found on https://www.geomar.de/en/research/fb1/fb1-me/research-topics/climate-modelling/kcms.

In FOCI, we have implemented with respect to the KCM updated versions of both the Atmosphere Model (ECHAM6.3.04, Müller et al., 2018), the coupling software (OASIS3-MCT, Valcke, 2013) and the ocean model (NEMO3.6, Madec, 2016). Additionally, FOCI optionally contains an atmospheric chemistry module (Schultz et al., 2018) and an ocean biogeochemistry module (Kriest, 2017) which in combination will enable our model system to close the carbon cycle.

A further novelty is the use of high-resolution ocean nesting in a climate model, based on the Adaptive Grid Refinement in

Fortran (AGRIF, Debreu et al. (2008)). The regionally refined ocean nest interacts with the global ocean by a two-way-nesting





technique. This key development to study high-resolution and mesoscale processes at multi-decadal timescales in forced global ocean model configurations at still reasonable computational costs (e.g., Biastoch et al., 2015; Durgadoo et al., 2013; Biastoch et al., 2009; Böning et al., 2016; Schwarzkopf et al., 2019) has now been extended to include coupling with the atmosphere. The coupled-nested climate model allows both 1) to explicitly include and simulate the atmospheric feedback in these high-

resolution nested configurations, and 2) to study the role of high-resolution features, such as mesoscale ocean eddies, onto the atmosphere and the climate system as a whole. In terms of computing costs and resources (Tab. 1), it has to be highlighted that a compromise was made which resolves the ocean at 1/10° and still yields an affordable coupled model with an atmosphere of currently ∼1.8° resolution. Note that at these resolutions the costs of the nested ocean model are much higher than those of the atmosphere model. Depending on the scientific topic, both ocean nesting and atmospheric chemistry can be used, and

run separately or concurrently within FOCI. One example for the application of FOCI with ocean nesting and atmospheric chemistry is the investigation of the shift of the Southern Hemisphere westerlies on the Agulhas leakage. The wind shift has been partly attributed to increasing greenhouse gases (GHGs) and the decline of stratospheric ozone (e.g., Perlwitz, 2011). The relative importance of GHGs and ozone depleting substances can now be studied explicitly in one consistent model system.

In this paper, we provide a technical description of the model components and configurations, as well as the coupling between

the components. In addition, we validate the model mean state and variability using a millennial pre-industrial control simulation and shorter multi-centennial nested control simulations, as well as a small ensemble of historical simulations covering the period from 1850 to 2013 with and without interactive atmospheric chemistry.

Section 2 provides technical details on FOCI's model components. Section 3 provides an evaluation of the mean state using different configurations of FOCI. The representation of selected global variability patterns in FOCI is discussed in section 4,

and a summary as well as an outlook on future model developments and implementations of FOCI is provided in section 5.

## 2 Model setup

FOCI is the successor of the Kiel Climate Model (KCM, Park et al., 2009) with updated model components and additional options that were not available in KCM. Technically, the current version of FOCI builds upon the build and runtime environment of MPI-ESM which has been adjusted to meet FOCI's requirements. As GEOMAR has recently joined the development of the

ESM-Tools (https://www.esm-tools.net), FOCI will switch to the ESM-Tools[1] runtime and compile environment in the future. In the following sections a short overview on the FOCI components is provided.

### 2.1 Atmosphere Model

FOCI uses the latest release of ECHAM (version 6.3.04) which is an improved version of ECHAM described in Stevens et al. (2013), and forms the atmospheric component of the Max Planck Institute's Earth System Model (MPI-ESM) contribution

to CMIP6 (Müller et al., 2018). ECHAM6.3 builds upon the previous version ECHAM6.1 used for CMIP5 of MPI-ESM. A detailed description of the model physics and changes from previous versions are described in (Müller et al., 2018). Here,

---

[1]A common runtime and compile environment developed at the Alfred Wegener Institute, Helmholtz Centre for Polar and Marine Research (AWI).





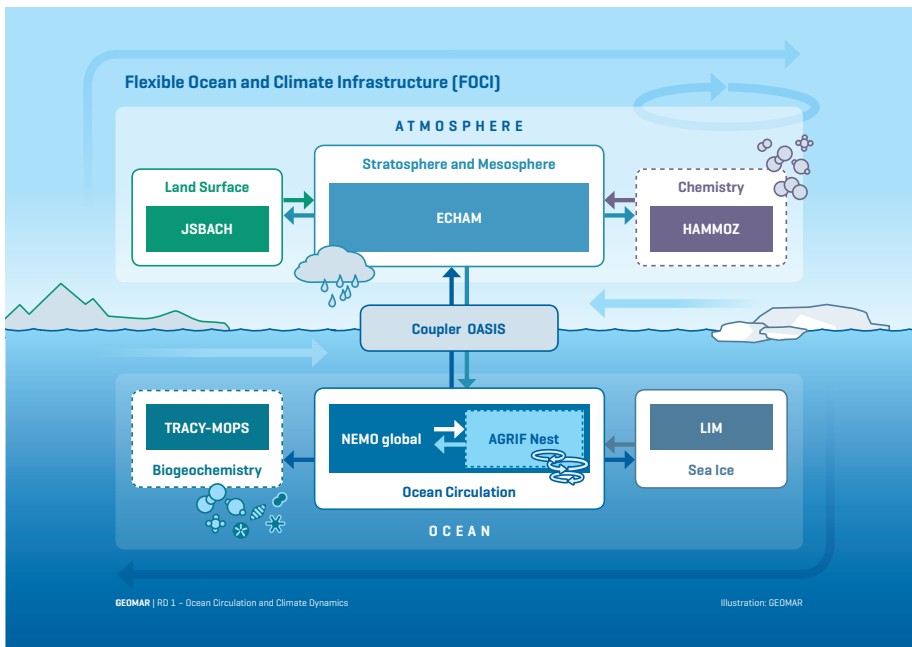

**Figure 1.** Schematic of the modeling system FOCI. Dashed boxes indicate optional modules.

we provide a short overview on the representation of the most important atmospheric processes in ECHAM6, based on the description provided in Stevens et al. (2013); Mauritsen (2019).

FOCI's default atmospheric component ECHAM6.3 dynamical core is spectrally resolved with a truncation wavenumber of 63 ($\approx 200\,km$). The model's physics are represented on a Gaussian grid in the horizontal, approximately 1.8° by 1.8°, and 95 vertical hybrid sigma-pressure levels with a model lid at 0.01 hPa. The vertical levels are distributed such that the stratosphere is resolved by about 1-2 km across the full vertical extent. The vertical resolution as well as the treatment of orographic gravity waves following Palmer et al. (1986) and Miller et al. (1989) and non-orographic gravity waves (Hines, 1990a, b) lead to an internal generation of a Quasi-Biennial Oscillation (QBO). FOCI's default settings with T63L95 corresponds to the MPI-ESM-MR version (Giorgetta et al., 2013), whereas the MPI-ESM-HR uses T127L95 (Müller et al., 2018).

The adiabatic core of ECHAM6 consists of a mixed finite-difference/spectral discretization of the primitive equations (Stevens et al., 2013). The convective parameterization is based on Tiedtke (1989) with modifications for penetrative convection according to Nordeng (1994). The radiative transfer calculation is updated every 2 hours using the rapid radiation transfer suite of models as optimized for general circulation modeling studies (RRTM-G, Iacono et al. (2008)) using 16 bands between 3333 nm and 1 mm in the longwave and 14 bands between 200 nm and 12195 nm in the shortwave spectrum. Trace gas concentrations are specified in ECHAM6, with the exception of water vapor, which is treated prognostically. Temporal variations in solar radiation are treated independently for each of the 14 shortwave radiation bands, and follow the CMIP6 recommendations provided by the SOLARIS-HEPPA project (Matthes et al., 2017) and http://solarisheppa.geomar.de/cmip6. This version of ECHAM6 uses the simple plume implementation of the second version (v2) of the Max Planck Institute Aerosol Climatology,





(MACv2-SP, Stevens et al. (2017)) to describe the effects of tropospheric aerosols. In the stratosphere, only sulfate aerosols resulting from volcanic eruptions are considered (Schmidt et al., 2013). The data set of volcanic forcing for the historic period from 1850 to 1999 is an extended version of the Pinatubo aerosol data set developed by Stenchikov et al. (1998), and is described in detail in Schmidt et al. (2013). Background aerosols are neglected.

Following the guidelines given in Mauritsen et al. (2012) we slightly retuned ECHAM6 to produce a global mean climate that is within the range of global observational estimates. With respect to the default settings used in MPI-ESM we slightly increased a parameter (cmfctop) that controls the convective mass-flux above the level of non-buoyancy since it has been shown to have a significant impact on the global mean temperature (Mauritsen et al., 2012).

FOCI is currently configured to run two versions of ECHAM6: The standard (physical) version described above, and the
ECHAM6-HAMMOZ version (Schultz et al., 2018). ECHAM6-HAMMOZ technically allows to run both the Hamburg Aerosol Model (HAM) and version 3 of the Model for Ozone and Related Chemical Tracers (MOZART3 or MOZ) separately or simultaneously. ECHAM6-HAMMOZ has currently two sets of chemical equations configured. FOCI uses the set of reactions used in ECHAM5-HAMMONIA (Schmidt et al., 2006), focusing on stratospheric chemistry. It uses 52 chemical tracers taking into account 185 reactions with 50 chemical reactions for photolysis-related reactions. Another computationally more expensive
set of chemical reactions describes both tropospheric and stratospheric chemistry in more detail with more than 600 reaction equations (described in detail in Schultz et al. (2018)). Schultz et al. (2018) validated in particular the tropospheric chemistry using nudging within ECHAM6. The ECHAM6-HAMMOZ configuration implemented in FOCI does not use nudging.

The lower atmospheric boundary conditions over land are simulated by JSBACH version 3 (Brovkin et al., 2009; Reick et al., 2013), an integral component of ECHAM6 to simulate biogeophysical and biogeochemical processes on the land surface. In
the current setup, JSBACH runs in the DYNVEG (DYNamic VEGetation) mode, which interactively simulates albedo, soil moisture, snow cover, leaf area, and vegetation distribution. The vegetation distribution is represented by PFTs (Plant Functional Types), in order to place and shift vegetation according to bioclimatic limits (e.g. Groner et al. (2018)). Additionally, wind break and fire act as disturbance processes for vegetation and the sub-model YASSO (Goll et al., 2015) determines the decomposition of carbon in the biosphere.

**2.2 Ocean/Sea-ice Model**

The oceanic component of FOCI is build on the NEMO code version 3.6 (Nucleus for European Modelling of the Ocean; Madec, 2016). The primitive equations for describing the dynamic-thermodynamic state of the ocean are discretized on a structured Arakawa (1966) C-grid. From the southern limitation of the model grid at 77°S to 20°N the coordinate lines follow a geographical Mercator grid. To avoid too small grid cells resulting from poleward convergence and to overcome the singularity
at the North Pole, north of 20° N, the coordinate lines deviate from Mercator, forming a tripolar grid (Madec and Imbard, 1996) extending to 90° N with two northern poles over Canada and Russia.

The nominal resolution of the first configuration is 1/2° (ORCA05), ranging from 55.6 km at the equator to 12.6 km in the Arctic Ocean. The global average resolution is 2/3 of this nominal resolution, hence 38.8 km. In the vertical, the water column is discretized on geopotential z-levels. The standard configuration uses 46 z-levels, ranging from 6 m at the surface to 250





m in the deep ocean. An important feature is that the bottom grid cells are allowed to be partially filled, ensuring realistic bathymetric gradients to adequately represent flow over $f/H$ contours ($f$ being the Coriolis Parameter and $H$ the water depth). Together with a momentum advection scheme conserving both energy and enstrophy (EEN, Arakawa and Hsu, 1990), this leads to an improved representation of the large-scale horizontal flow field (Barnier et al., 2006). For tracer advection, a 2-step

flux-corrected transport, total variance dissipation scheme (TVD, Zalesak, 1979) is used, ensuring positive-definite values. We use a linearized filtered free surface. The time step for the baroclinic components is 1800 s.

Viscosity is applied through a bi-Laplacian operator (nominal values of $-12 \times 10^{11}\ m^4 s^{-1}$, scaled by the grid width), tracer diffusion is aligned along isopycnals, with isoneutral diffusion of $600\ m^2 s^{-1}$. Horizontal sidewall boundary conditions are formulated as free-slip conditions, in the vertical a quadratic bottom friction is applied. In the upper ocean, a turbulent kinetic

energy (TKE) mixed layer model (Blanke and Delecluse, 1993) diagnoses the depth of the mixed layer and increases vertical mixing for unstable water columns. This includes the handling of deep convection in formation regions of deep and bottom waters.

With the ORCA05 grid the model is not able to resolve the generation processes of mesoscale eddies. Therefore, an eddy parameterization scheme following (Gent and McWilliams, 1990) is applied, with actual values calculated from the temporally

and horizontally varying growth of the baroclinic instability up to a maximum of $2000\ m^2 s^{-1}$ (Treguier et al., 1997). This configuration thus has the character of a coarse-resolution model yielding a rather laminar oceanic flow field (Biastoch et al., 2008a).

The dynamic-thermodynamic sea-ice component is based on the Louvain-la-Neuve sea Ice Model version 2 (LIM2) first described in Fichefet and Morales Maqueda (1997). The version used here is part of NEMO3.6 and comprises settings that

have proven beneficial for the application of the nesting capability of FOCI (see below). LIM2 is run with the viscous-plastic rheology (Hibler III, 1979) using an ice-strength parameter $P^*$ of 15,000 Nm$^{-2}$ and a single ice thickness category with a lead closing parameter $h_0$ of 0.3 m and 0.6 m for the northern and southern hemisphere, respectively. Uniform ice and snow thickness distributions ranging from 5 cm to twice their mean thickness are assumed for the calculation of heat fluxes however. While LIM2 is less comprehensive than state-of-the-art sea-ice models, it still yields a proper representation of sea ice for the

purpose of global climate simulations and it runs stably when nesting is applied.

## 2.3 Coupling

Ocean and atmosphere are coupled without flux adjustment via the OASIS3-MCT coupler (Valcke, 2013). It performs the exchange of momentum, sea-ice properties and heat and freshwater fluxes between both components. It further handles the necessary grid transformations between the Gaussian grid in the atmosphere and the oceanic tripolar ORCA05 grid. Therefore

conservative remapping functions of the spherical coordinate remapping and interpolation package are used in the OASIS3-MCT coupler (SCRIP, Jones, 1999). First-order conservative remapping is applied on all scalar quantities and ocean currents, but bicubic remapping (on a 16-point stencil for Gaussian reduced grids) on wind stress. All flux calculations are performed in the atmospheric part of FOCI. Since the ~1.8° atmospheric grid is considerably coarser than the oceanic one (1/2° or even 1/10° when nesting is applied), fluxes are significantly smoothed in space. Therefore the effects of atmosphere-ocean





| | | ORCA05 | ORCA12 | INALT10X | VIKING10 |
|---|---|---|---|---|---|
| Number of ocean grid points | | 511×722×46 =16.971.332 | 4322×3059×46 =608.165.908 | 1404×924×46 +511×722×46 =76.646.948 | 868×884×46 +511×722×46 =52.267.684 |
| Ocean only | CPU hours / model year | 250 | 37600 | 3200 | 2200 |
| | total CPU | 336 | 2688 | 1080 | 912 |
| | 100 yr simulation (hours) | 74 | 1399 | 296 | 241 |
| FOCI Coupled | CPU hours / model year | 700 | *45000* | 4100 | 3000 |
| | total CPU | 936 | *3200* | 1392 | 1224 |
| | 100 yr simulation (hours) | 75 | *1406* | 295 | 245 |
| FOCI Coupled chem | CPU hours / model year | 2736 | - | 7200 | - |
| | total CPU | 1368 | - | 1824 | - |
| | 100 yr simulation (hours) | 200 | - | 400 | - |

**Table 1.** High-performance computing resources for various configurations of the FOCI model system. These values are based on a CRAY XC30 machine at ZIB Berlin, part of the North-German Supercomputing Alliance HLRN, and estimates of non-existing configurations in italic. All coupled experiments use a T63L95 atmosphere resolution. The total number of ocean grid points is depicted in the first row for two global (ORCA05 1/2°, ORCA12 1/12°) and two nested configurations at 1/10° in a 1/2° host grid (INALT10X and VIKING10). Rows 2 to 4 specify resources for uncoupled experiments and coupled experiments without and with atmospheric chemistry, respectively, for all resolutions.

feedbacks are compromised. It is planned to resolve this issue in a future version of FOCI. River runoff is calculated by a horizontal discharge model (Hagemann and Dümenil Gates, 2003) and passed to the ocean as a combined freshwater flux by the coupler at the nearest atmospheric grid cell with sufficient ocean coverage underneath.

The coupling frequency is every 3 hours to adequately capture the diurnal cycle. It can be modified at any time to the appropriate
needs. The use of massive parallelization in all model components yields a throughput of up to 30 model years per day on 39 nodes comprised of 24 cores each, for the standard FOCI configuration (T63L95 and ORCA05), see Tab. 1.

### 2.4 Ocean Nesting Capability

In FOCI, the ocean grid resolution can be refined in a specified region by using a two-way-nesting approach for the ocean component. The so-called nest is embedded in the global host grid applying the Adaptive Grid Refinement in Fortran (AGRIF,
Debreu et al., 2008). It is executed in parallel being forced at its boundaries by the global ocean and atmosphere state and frequently feeds back its fine-scale 3-D state to update the solution on the host grid of the ocean model. Using ORCA05 as basis for the host grid, resolution enhances by a factor of 5 from 1/2° to 1/10° in the nest necessitating a smaller baroclinic time step of 600 s resulting from a temporal refinement factor of 3 to ensure numeric stability.

With FOCI, this method is applied to a fully coupled global climate model for the first time. It provides an opportunity to
systematically study the effect of mesoscale ocean dynamics in a given region by performing the same experiments without



(default version) and with grid refinement. The first two ocean nests implemented and tested in FOCI cover the South Atlantic and western Indian Ocean (Fig. 2 left, FOCI_INALT10X) and the North Atlantic between 30°N and 85°N (Fig. 2 right, FOCI_VIKING10). Both nest versions, for the North and the South Atlantic, currently run in individual setups and not simultaneously.

5 The ocean component of FOCI_INALT10X is part of the INALT-family (Schwarzkopf et al., 2019). It builds a successor of INALT01 (Durgadoo et al., 2013) which has been verified for its use in studying the Agulhas region in forced ocean-only experiments (Loveday et al., 2014; Biastoch et al., 2015; Lübbecke et al., 2015). FOCI_VIKING10 is motivated by the VIKING20 model of Böning et al. (2016) and is used to expand these studies to the impact of Greenland meltwater on the coupled climate system on multi-decadal to centennial timescales.

In the coupled-nested configurations of FOCI, the global host and regional nest receive the same atmospheric surface forcing.

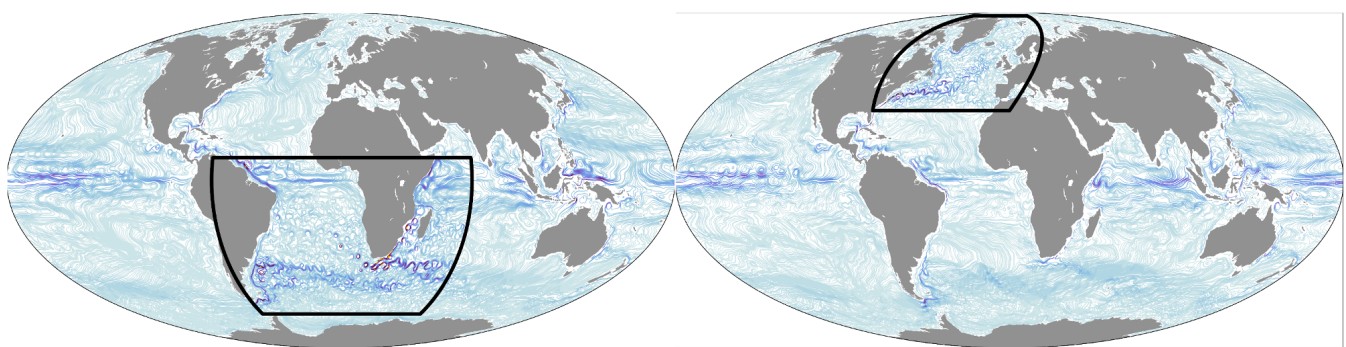

**Figure 2.** Snapshots of upper ocean current speed (upper 100m averaged) for two separate ocean nest configurations, where ocean resolution refines from 1/2° to 1/10° in the highlighted regions. Left: INALT10X, South Atlantic nest between 9°N - 63°S, 70°W - 70°E, Right: VIKING10, North Atlantic nest between 30°N - 85°N.

The host is actually coupled to the atmosphere in the traditional sense via the coupler while the forcing for the nest is provided as follows. At the atmosphere-ocean coupling time step, here every 3 hours, the coupler provides the atmospheric fields to the host thereby also conducting the necessary remapping between atmosphere and ocean grid. The atmospheric fluxes are additionally saved by the coupler on their native grid. These forcing fields are then read by the nest and remapped in a very similar way as done by the coupler for the host. In the ocean component, the host is updated with the complete 3-D ocean state of the nest prior to every coupling time step with the atmosphere. This two-way-nesting (Debreu et al., 2008) ensures that surface ocean conditions, as simulated by the finer grid, are available for the surface flux calculations, though spatially smoothed.

The coupled-nested configuration has a five-times higher computational demand compared to the default version of FOCI due to the increase in ocean grid points and the associated time step refinement of factor 3. For example, FOCI_INALT10X features about 60 million grid points whereas ORCA05 has only 17 million grid points, with both comprising 46 vertical levels. Together with a refined timestep, this yields an expense increase by a factor 5 compared to the default FOCI configuration without nest. With a computational throughput of 10 model years per day (on 52 nodes) multi-decadal nested simulations are well operable (see Tab. 1).





Ocean model parameters that scale with either spatial or temporal resolution need to be adjusted for the nest while parameters for the host remain unchanged with respect to the default configuration without nest. Noteworthy parameters are bi-Laplacian viscosity, changed from $-60 \times 10^{10}\, m^4 s^{-1}$ to $-2.4 \times 10^{10}\, m^4 s^{-1}$, isoneutral diffusion from $600\, m^2 s^{-1}$ to $120\, m^2 s^{-1}$ and eventually the horizontal sidewall boundary conditions. The latter are changed to no-slip conditions in case of

FOCI_INALT10X but kept at free slip for FOCI_VIKING10. This choice depends on the validation of the simulated boundary currents.

### 2.5  Marine Biogeochemistry

The marine biogeochemistry model MOPS (Model of Oceanic Pelagic Stoichiometry) is available in FOCI. This model is

based on the cycling of nutrients (phosphate and nitrate), phytoplankton, zooplankton, detritus, and dissolved inorganic matter (Kriest and Oschlies, 2015). MOPS has been coupled to the Transport Matrix Method (Khatiwala, 2007), with monthly mean transport matrices (TMs), wind speed, temperature and salinity (for air-sea gas exchange), and is calibrated against global climatologies of observed macronutrients and oxygen (Kriest, 2017; Kriest et al., 2017), yielding an objectively optimized set of biogeochemical parameters for a given circulation and metric, and an uncertainty envelope for the optimized biogeochemical

parameters. As the parameters can depend on the circulation of the physical ocean model, parameter optimizations have been performed with five different TMs. The results are currently being assessed with respect to the impact of circulations on optimal model parameters (Kriest et al., in prep). The carbon cycle was coupled to MOPS by applying a constant stoichiometry of C:P=117 and the marine carbonate system in the upper ocean is described following Orr and Epitalon (2015) with the difference that carbonate chemistry is only computed for the ocean surface layer and therefore pressure corrections are not included.

The air-sea gas exchange for $CO_2$ and oxygen used in the carbonate chemistry description and MOPS are based on Orr et al. (2017) and the parameterisation for the calcium carbonate pump follows (Schmittner et al., 2008).

In order to achieve a low enough uncertainty in tracer mass to account for anthropogenic signals in atmosphere-ocean $CO_2$ fluxes, the non-linearized free surface setting for NEMO is necessary (an uncertainty of $10^{-9}$ in the mass of a passive tracer is being achieved). The evaluation of FOCI as an Earth system model with a fully coupled carbon cycle is still in progress and

subject of a future publication.

### 2.6  Experimental Setup

Table 2 provides an overview on the FOCI model simulations presented in this paper. The 1500 year long FOCI reference simulation (FOCI-piCtl) is initialized using the PHC2.1 climatology (Steele et al., 2001) for temperature and salinity in the ocean.

Sea-ice is initialized using a restart file from an uncoupled ocean-only hindcast simulation (Behrens et al., 2013, experiment "WEAK 05"). The simulated sea-ice state of December 31st, 1993 was found to resemble qualitatively well the observed mean state prior to the onset of the recent ice loss.

For JSBACH, carbon and nitrogen pools as well as land cover fractions are initialized using restart data from a multi-millenial



| ID | years (#) | description |
|---|---|---|
| FOCI-piCtl | 0001-1500 (1500) | pre-industrial control run under 1850 climate conditions initialized from an ocean at rest and Levitus 1998 data for temperature and salinity |
| FOCI-hist[1,2,3] | 1850-2013 (3x164) | historical simulations with observed CMIP6 external forcing restarted from year 1030, 1041 and 1499 from FOCI-piCtl, respectively. |
| FOCI-chem[1,2,3] | 1950/52/54-2013 (64/62/60) | historical simulations with interactive atmospheric chemistry (HAMMOZ) initialized from year 1950, 1952 and 1954 from a FOCI historical type simulation |
| FOCI_VIKING10-piCtl | 1500-1650 (150) | pre-industrial simulation under 1850 climate conditions initialized from end of FOC-piCtl using the VIKING10 ocean nest |
| FOCI_INALT10X-piCtl | 1500-1650 (150) | pre-industrial simulation under 1850 climate conditions initialized from end of FOC-piCtl using the INALT10X ocean nest |

**Table 2.** Overview of FOCI simulations.

MPI-ESM Holocene simulation provided by the MPI that incorporates land use changes from 0 to 1850 AD. This ensures that especially the slow carbon pools and the biosphere are in a well-balanced and spun-up state. Interactive vegetation is used with land use changes corresponding to 1850s values. Nevertheless an adjustment process of the land vegetation will take place during the first 500 to 1000 years of the FOCI-piCtl presented due to changes in the temperature and precipitation distribution

which stem from the replacement of the ocean model in FOCI with respect to MPI-ESM. The control experiment runs for 1500 years using pre-industrial external forcing from 1850, and if not stated otherwise, the last 500 years from this run are used to evaluate the model performance.

In total 6 historical CMIP-type simulations have been performed for the purpose of the present study. Three simulations without interactive chemistry in the atmosphere (FOCI-hist[1,2,3]), covering the full historical period from 1850 to 2013[2]. These

simulations were initialized using model year 1030, 1041 and 1499 from FOCI-piCtl. Restart years were chosen with respect to their initial sea-ice cover in the Southern Hemisphere (SH) (FOCI-hist[1,2]) due to a SH sea-ice bias in the current version of FOCI as discussed in more detail in section 3.3. The simulation labeled FOCI-hist3 starts from the final year of the control simulation and serves as a reference simulation for nested FOCI historical simulations that will be presented in forthcoming papers. Three additional historical experiments with the FOCI standard configuration were conducted using the interactive

atmospheric chemistry module (FOCI-chem[1,2,3]). These simulations all branch off from a FOCI historical run, the latter restarted from the reference FOCI-piCtl run at year 1000.

The nested experiments branch off from the FOCI-piCtl after 1500 model years, serving as a spin-up, and then integrated forward under 1850 pre-industrial conditions for 150 years (Tab. 2). Further FOCI configurations such as FOCI_INALT10X-chem as well as future scenario runs with and without nests are established as part of ongoing research projects.

---

[2]Due to technical reasons the FOCI version presented here was not able to cover the full CMIP6 historical period until 2014.



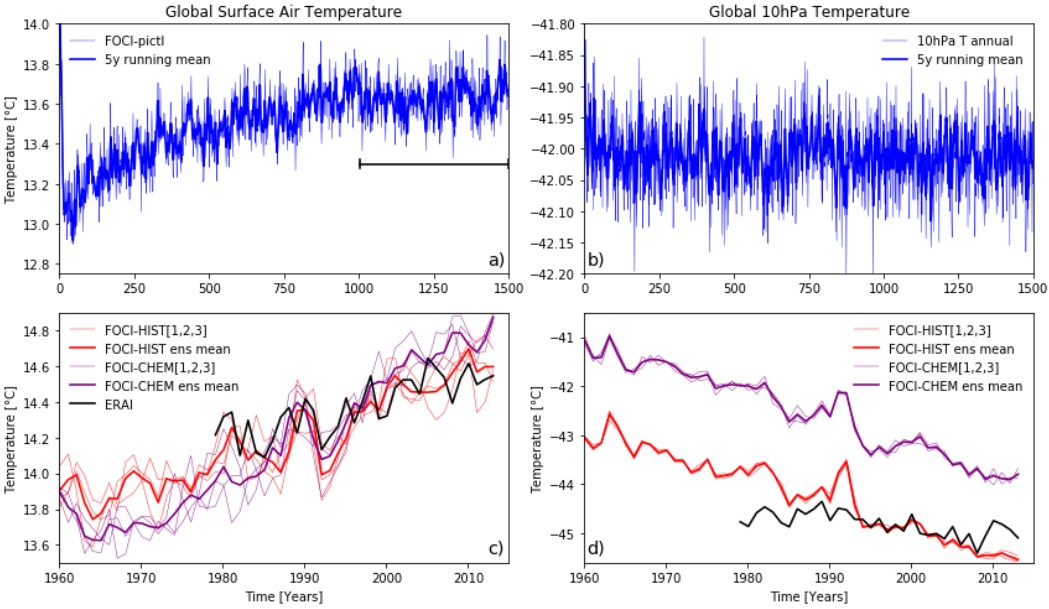

**Figure 3.** (a) Global surface air temperature from FOCI-piCtl, (b) global 10 hPa temperature from FOCI-piCtl, and (c) FOCI-hist from 1950 through 2013 and (d) FOCI-chem. The black bar in (a) highlights year 1000 to 1500 which are used to evaluate the mean state of FOCI throughout this publication. For the historical period in (c,d) the ERA-interim reanalysis temperature is included as observational estimate.

## 3 Mean State

### 3.1 Atmosphere Mean State

Figure 3a shows the globally averaged surface air temperature (SAT) over the whole reference simulation. After an initial drop, SAT stabilizes around approximately 800 years at 13.6°C which is in excellent agreement with the global estimate of
pre-industrial SAT of approximately 13.5°C (e.g., Mauritsen et al. (2012)). The globally averaged temperatures in the strato-sphere at 10 hPa stabilize very fast at approximately -42°C (Figure 3b). In our set of six historical simulations with and without stratospheric chemistry (Figure 3c) the overall global temperature increase during the last 50 years as well as the dip in global mean temperature during the first part of the 1990s is captured very well by the ensemble means and might be related to the Pinatubo eruption. The slowdown in global warming during the so-called climate hiatus period starting in 1998 is very well
captured by the FOCI-hist ensemble. In general the SAT trends are larger in both FOCI ensemble means than in ERA-Interim for both periods and clearly show the global warming as compared to the negligible trends in the FOCI-pictl simulation (Figure 3c, Table 3). Noteworthy is the stronger SAT trend in the FOCI-chem ensemble as compared to the FOCI-hist ensemble.

Similarly, the observed temperature decrease in the stratosphere during the last decades which is due to the combined effect
of increasing greenhouse gases and reduced ozone concentrations in the stratosphere is simulated in all FOCI historical model

| configuration | years | SAT | 10hPa T |
|---|---|---|---|
| FOCI-pictl | 1000-1499 | +0.056K | +0.002K |
| FOCI-hist | 1980-2013 | +0.611K | -2.047K |
| FOCI-chem | 1980-2013 | +0.959K | -2.033K |
| ERAI | 1980-2013 | +0.415K | -0.530K |
| FOCI-hist | 1994-2008 | +0.340K | -0.628K |
| FOCI-chem | 1994-2008 | +0.527K | -0.483K |
| ERAI | 1994-2008 | +0.238K | -0.491K |

**Table 3.** Trends of SAT and 10hPa temperature in FOCI-pictl as well as FOCI-hist and FOCI-chem ensemble means compared to ERA-Interim data for different time periods.

experiments (Figure 3d, Table 3). An offset of approximately 2 K can be found in the FOCI-chem ensemble as compared to the FOCI-hist ensemble and the ERA-interim data, which will be discussed in more detail below. The simulated stratospheric temperature trends by FOCI are larger than those from ERA-Interim over the 1980 through 2013 period. FOCI-chem agrees better with ERA-Interim than FOCI-hist in the period of flattening of stratospheric temperature trends between 1994 and 2008
(Table 3) indicating that a more realistic ozone field from the interactive stratospheric chemistry is important for realistic temperature and circulation evolution. Note however, that ERA-Interim is only one estimate of the observed state and may also contain uncertainties.

Another key parameter for a coupled global climate model is the radiation balance at the top of the atmosphere (TOA). Due to model errors, numerical approximations or the parameterization of sub-grid scale processes such as convection, energy is not
fully conserved in coupled climate models. The current version of FOCI shows a radiation imbalance of +0.7 W/m$^2$ which is in the range of imbalance we see in other climate models Mauritsen et al. (2012); CMIP3, Lucarini and Ragone (2011); CMIP5, Hobbs et al. (2016)), larger than the values achieved in the latest version of MPI-ESM (Müller et al., 2018), but close to the current observed imbalance of +0.5 W/m$^2$ (Roemmich et al., 2015). Considering that the evolution of the SAT is relatively stable (e.g., Figure 3c), we expect a gradual warming of the ocean (see section 3.2). Please note that the vertical distribution of
LW and SW heating rates compares well between FOCI-hist, FOCI-chem and MERRA reanalysis (Fig. A5). However, the net heating rates in FOCI show in contrast to MERRA a slight imbalance mainly in the upper stratosphere and lower mesosphere.

In the following we present the differences between the last 500 years of the FOCI-piCtl simulation and 34 years of the ERA-interim reanalyses to evaluate the model's overall performance. Please note that the background state of the reference
simulation differs of course from the historical observed period. The global bias in SAT pattern (Fig. 4a, and Fig. A1 for the seasonal differences) is common to coupled climate models. The SST bias in ice-free regions in the North Atlantic (Fig. 13) reaches up to -7°C in the annual mean and will be discussed in more detail in sections 3.2 and 3.4. The large positive annual mean SAT bias of up to +6.5°C in the Weddel Sea over the Southern Ocean at the edge of Antarctica is related to a pronounced



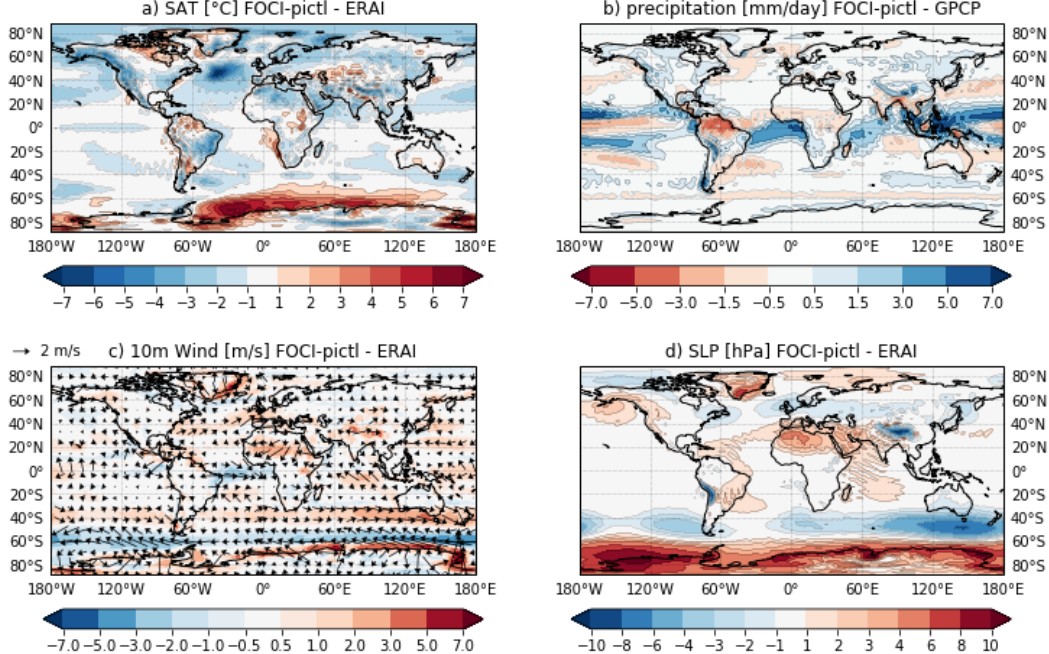

**Figure 4.** Annual mean biases for the FOCI-piCtl simulation for (a) surface air temperature in Kelvin (K), (b) precipitation in millimeter per day (mm/d), (c) 10 m wind in meter per second (m/s), and (d) 500 hPa geopotential height (Z500) in hPa with respect to ERA-Interim (a,c,d, 1980-2013) or the GPCP dataset (b, 1980-2013). Please note reversed color scale in b) to highlight dry (wet) areas in shades of red (blue) compared to GPCP.

negative bias in sea ice in the same region, and will be discussed in more detail in section 3.3.

Global annual mean precipitation shows significant biases with respect to the Global Precipitation Climatology Project (GPCP Adler et al., 2018) in the tropics as depicted in Fig. 4b. It varies with season (Fig. A2). The pattern indicates too intense convective precipitation over the maritime continent, which might be influenced by the slight negative SST bias along the
5 equatorial Pacific which has also implications for FOCI's ability to simulate ENSO as discussed in section 4.5. A southward shift of the Intertropical Convergence Zone (ITCZ) is present in the tropical Atlantic in conjunction with a pronounced dry bias over northern Brazil peaking in boreal spring and summer (Figure A2). The basic structure of the tropical Atlantic precipitation bias pattern is already present in uncoupled ECHAM6 model experiments (Stevens et al., 2013) and is amplified through feedbacks in a coupled ocean-atmosphere setup (e.g., Richter and Xie, 2008; Wahl et al., 2009). Additionally, other ECHAM-
10 based coupled models, e.g. MPI-ESM (Giorgetta et al., 2013; Stevens et al., 2013; Müller et al., 2018) and the Kiel Climate Model (Park et al., 2009) show very similar precipitation bias pattern. Global mean precipitation stabilizes at 2.85 mm/day which is about 5% higher than the observational estimate of 2.69 mm/day derived from the GPCP dataset.

Figure 4c shows the annual mean 10 m wind bias with respect to the ERA-Interim climatology. A prominent feature in the tropical Atlantic are weaker trade winds, in agreement with the precipitation bias. The strongest wind bias in the tropical



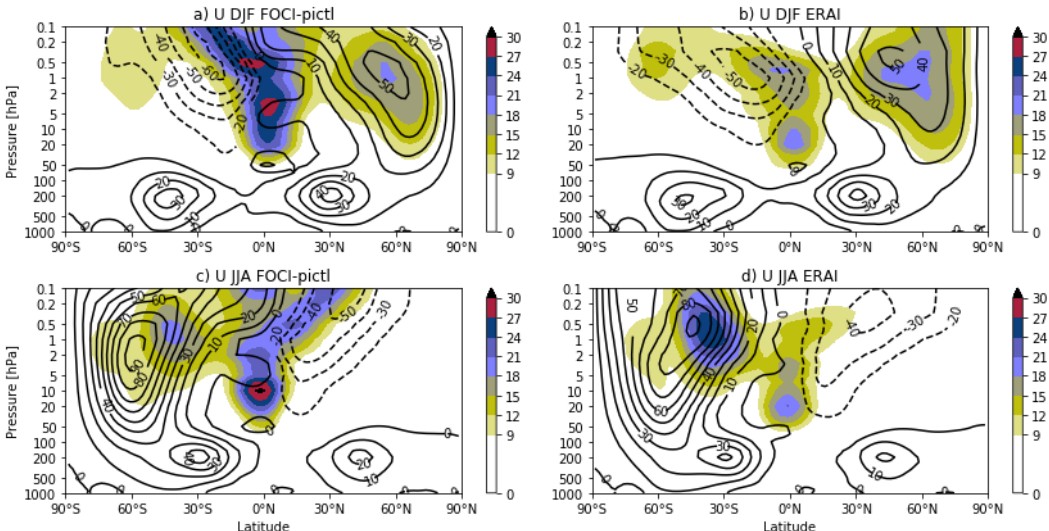

**Figure 5.** Zonal mean zonal wind climatology from FOCI-piCtl for (a) DJF, and (c) JJA, and ERA-Interim reanalysis climatology (1979 to 2017) for (b) DJF, and (d) JJA; contour interval 10 m/s, shading indicates the zonal mean zonal wind standard deviation in m/s.

Atlantic peaks in boreal spring (not shown) and is related to a southward shift of the ITCZ and a dry bias over northern Brazil. This bias is a common feature of climate models (e.g., Richter and Xie, 2008; Chang et al., 2008; Richter et al., 2012) and has its source in the atmosphere model since it is already present in ECHAM6 atmosphere-only configuration with prescribed SSTs (Stevens et al., 2013). Another strong wind bias occurs in the Southern Ocean, where the westerlies in FOCI-piCtl are shifted

5  towards the equator, compared to recent reanalysis data. The equatorward shift persists in the FOCI-hist simulations with a pure shift in the Pacific sector, a slight equatorward shift but strong overestimation in the Atlantic sector and combination of both in the Indian sector. The wind bias agrees with the sea level pressure (SLP) bias pattern (Figure 4d, see Fig. A3 for the seasonal SLP bias pattern). The equatorward shift of the westerlies might contribute to the weak representation of the Antarctic Circumpolar Current (ACC, Figure 11), but is most likely not the main reason as discussed in more detail in section 3.2.

The vertical structure of the atmosphere in FOCI with respect to ERA-Interim is presented in Figs. 5 and 6 for zonal mean wind and zonal temperature during boreal winter and summer. The polar night jet in DJF with its maximum of about 50 m/s and its variability of about 18 m/s in the mid-latitude upper stratosphere/lower mesosphere as well as the equatorward tilt is captured very well in FOCI (Fig. 5). Zonal mean zonal winds in the mid to lower mid-latitude stratosphere are slightly

15  stronger in FOCI. The easterly jet in the summer hemisphere is about 10 m/s stronger in FOCI as compared to the reanalyses. The standard deviation in the tropical stratosphere and lower mesosphere due to the internally generated QBO in ECHAM is overestimated by FOCI and will be discussed in more detail in section 4.2. The tropospheric jets in DJF are well represented. Larger differences occur in JJA, where the polar night jet in FOCI shows a larger amplitude and a slightly shifted maximum at 60°S and 1-2 hPa, whereas the jet core in ERA-interim peaks more towards 40°S and 0.5-1 hPa. The overestimation of the



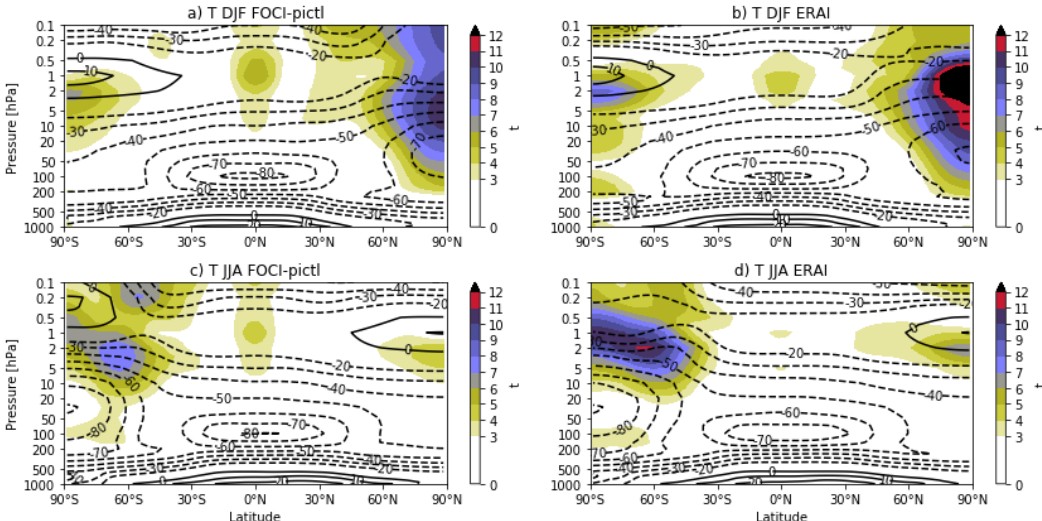

**Figure 6.** Zonal mean temperature climatology from FOCI-piCtl for (a) DJF, and (c) JJA, and ERA-Interim reanalysis climatology (1979 to 2017) for (b) DJF, and (d) JJA; contour interval 10°C, shading indicates the standard deviation of zonal mean temperature in °C.

zonal wind amplitude is also apparent for the easterly jet in the summer hemisphere and slightly stronger tropospheric jets. The maximum of the standard deviation in the winter hemisphere is weaker in FOCI and disconnected from the jet core in contrast to reanalyses. In the tropical stratosphere and mesosphere, the variability is consistent with DJF also stronger in FOCI. Similar to the zonal mean zonal wind, the zonal mean temperature climatology (6) is in general in good agreement with reanalyses data.

Except for the variability during NH and SH winters, which is significantly reduced in FOCI and results in a fewer number of sudden stratospheric warmings on the NH (section 4.3).

### 3.1.1 Interactive versus prescribed stratospheric chemistry

The performance of ECHAM6-HAMMOZ within FOCI can only be evaluated using simulations of the recent past, as very little information exists on the chemical composition of the atmosphere in pre-industrial times. Since the focus of this paper is

10 the introduction of FOCI in general and not the chemistry part of FOCI, we will limit the analysis of the impact of interactive versus prescribed chemistry to the performance of FOCI-chem with respect to total column ozone (TCO) observations as well as the vertical structure of zonal mean wind and temperature fields. A more detailed presentation of the performance of ECHAM6-HAMMOZ within FOCI will be subject of a forthcoming paper.

Numerous studies have highlighted the importance of interactive chemistry especially in the stratosphere and mesosphere to

15 properly represent stratosphere-troposphere coupling (e.g. Haase and Matthes, 2019) as well as solar influence on climate (e.g. Mitchell et al., 2015; Thiéblemont et al., 2015). The ozone hole as well as ozone chemistry play a very important role for the interaction with the large scale dynamics in the stratosphere (e.g. Lubis et al., 2017; Babington, 2018). In ECHAM6-HAMMOZ, a detailed set of reaction equations determines ozone concentrations in the atmosphere at every gridpoint based



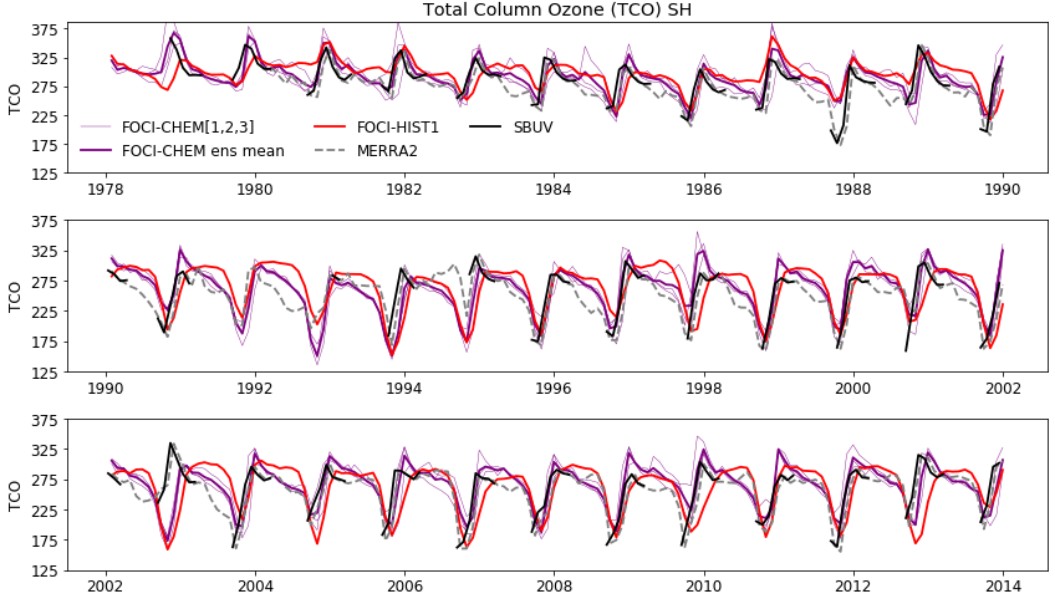

**Figure 7.** TCO time series from 1978 to 2013 from FOCI-hist (red), FOCI-chem (violet), MERRA2 reanalysis data (grey) and SBUV satellite observations (black) using zonal averages from 70° - 80°S. For details on the respective averaging period see text. TCO from FOCI-hist is derived from the ozone climatology used in FOCI-hist as ozone is prescribed in FOCI-hist.

on monthly estimates of surface emissions of ozone depleting substances (ODS) and variations in incoming solar radiation through changes in photolysis rates, while in standard ECHAM6 ozone is prescribed based on historical ozone concentrations taken from the CMIP6 database (Hegglin et al., 2018). Variations in the strength of incoming solar radiation therefore directly project onto ozone concentrations in FOCI-chem.

5   Figure 7 shows the development of total column ozone (TCO) in the southern hemisphere from 70°S to 80°S from 1978 through 2013 as simulated by FOCI-chem (violet) as well as FOCI-hist (red), MERRA2 reanalyses (grey) and SBUV satellite observations (Frith et al. (2014), black). We limit the analysis to 80°S as observational datasets either don't measure south of 80°S (SBUV) or only cover the period from October to March (Bodeker et al. (2005)[3]). The TCO from FOCI-hist hence represents the TCO calculated from the CMIP6 recommended and prescribed historical ozone dataset (Hegglin et al. (2018)) 10 used in FOCI's radiation scheme when no interactive chemistry is used. The differences in (total column) ozone between FOCI-hist and observations/reanalysis (Figure 7) are due to the origin of historical ozone concentrations in CMIP6 which are based on a multi-model mean of coupled chemistry-climate simulations. The amplitude and the timing of the ozone hole, with a maximum in September and October since the 1980's is very well captured in FOCI (see also Fig. A4). However, we do not expect the free-running FOCI model system to capture the observed interannual variability, such as the dynamically very 15 specific situation in 2002 and the resulting very weak ozone hole.

---

[3]an updated paper is currently in preparation http://www.bodekerscientific.com/data/total-column-ozone

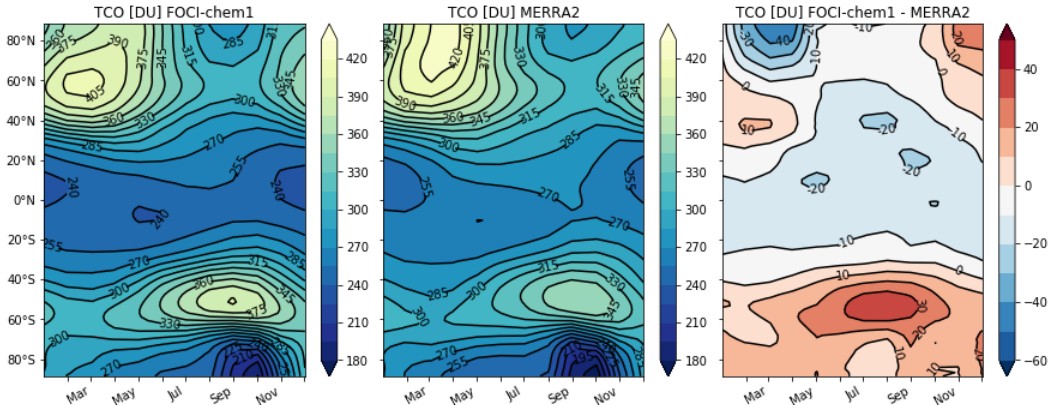

**Figure 8.** Zonal mean TCO climatology from (a) the FOCI-chem1 simulation, (b) the MERRA2 reanalysis (contour interval 15 Dobson Units (DUs)), and (c) the differences between (a) and (b) from 1980 to 2013, contour interval 10 DUs.

A global perspective of the seasonal cycle of TCO in FOCI with respect to MERRA2 is presented in Fig. 8. While TCO column ozone in the tropics is slightly weaker in FOCI than in observations we note the excessive ozone in the Southern Hemisphere mid-latitudes as well a too little ozone in boreal spring north of 60°N. Since ozone is photochemically produced in
the tropics and then transported away from its source region towards higher latitudes by the meridional overturning circulation, the Brewer-Dobson circulation (BDC) of the stratosphere, the deviations between FOCI and observations hint to a stronger BDC in the model, a common feature in current state-of-the-art chemistry-climate models Babington (2018). Small differences of about 10 DU occur in the area of the ozone hole in September and October. It's amplitude and timing is better captured in FOCI than most other CCMs Babington (2018). The discrepancy during boreal spring between FOCI and observations needs
further investigation and is beyond the scope of the current paper. The differences between FOCI and TCO observations are consistent for other observational datasets such as the NIWA-BS dataset (Bodeker et al. (2005)[4])

The impact of interactive chemistry on the zonal mean state is presented in Figure 9. As already shown in Fig. 3 the stratosphere is on average 2 K warmer in the FOCI-chem simulations. Similarly the mesosphere is between 2-4 K colder. The polar night
jet on the respective winter hemispheres is up to 3 m/s stronger with interactive chemistry. During NH winter the differences in zonal mean zonal wind extend from the stratosphere into the troposphere hinting to an impact of stratosphere-troposphere coupling. Large differences also occur in the tropical stratosphere and lower mesosphere which are related to the QBO as discussed in more detail in section 4.2. These differences are probably related to both differences in the ozone fields as well as dynamical differences between the FOCI simulations and are subject of a more detailed future study.

---

[4]see http://www.bodekerscientific.com/data/total-column-ozone for updates since 2005.



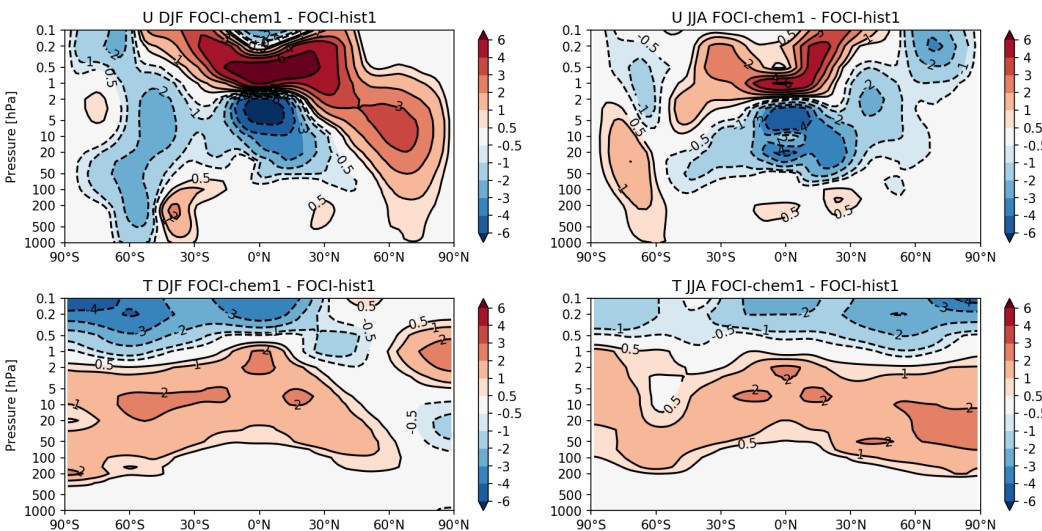

**Figure 9.** Differences between the FOCI-chem1 and the FOCI-hist1 simulations averaged from 1980-2013 in (a) DJF zonal mean zonal wind, and (b) JJA zonal mean zonal wind in m/s, (c) DJF zonal mean temperature, and (d) JJA zonal mean temperature in K. Please note the non-linear colorbars.

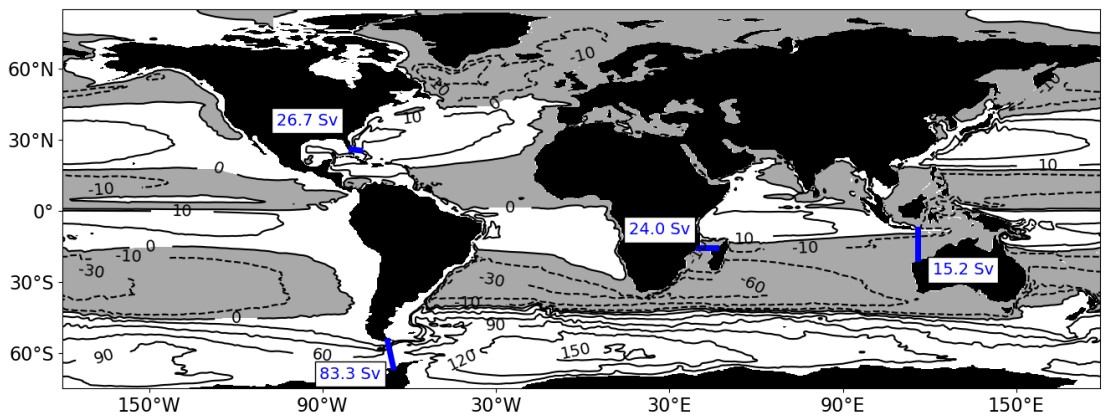

**Figure 10.** Time-mean barotropic stream function (in Sverdrup, Sv) derived from the last 500 yrs of the FOCI-piCtl simulation. White/grey shading indicates clockwise/anticlockwise rotation. The black lines represent the contours for 10 Sv as well as in 30 Sv intervals up to 150 Sv. The blue lines indicate selected transport sections (see Fig. 11) with mean transport values in the box aside.





## 3.2 Ocean Mean State

The time-mean barotropic stream function illustrates the representation of the large-scale oceanic regimes, mainly a result of the representation of the wind-driven circulation. Figure 10 shows that the anticyclonic subtropical gyres are well represented in FOCI. In the Northern Hemisphere, the corresponding western boundary currents (WBCs) are the Gulf Stream and the

Kuroshio. The cyclonic circulation in the subpolar North Atlantic has a strength of 35 Sv (Sverdrup; $1 \, \text{Sv} := 10^6 m^3 s^{-1}$), which is in the range of observations (Zantopp et al., 2017). In the Southern Hemisphere, WBCs off Brasil, South Africa (Agulhas) and Australia are well represented. The subtropical gyres of the Indian Ocean and South Atlantic are connected, forming a so-called supergyre, as observed by Ridgway and Dunn (2007) using World Ocean Database and Argo profiles. The 'Agulhas leakage', representing the net interoceanic transport from the Indian into the Atlantic ocean through the Agulhas

Current system is at the order of 30 Sv, which is twice the amount of observational estimates (Richardson, 2007) and a typical problem of coarse-resolution Ocean GCMs (Biastoch et al., 2008b).

The Antarctic Circumpolar Current (ACC) shows a recirculation of 150 Sv in the Atlantic sector in line with observations (Evans et al., 2017), while its overall strength, i.e. defined as the volume transport through the Drake Passage, is at the lower bound of observational estimates.

Transport time series of some prominent passages, derived from the barotropic stream function are shown in Fig. 11. The transport through the Florida Strait is a well-established and robust value, continuously observed by measurements of the integral transport across telephone cables (Meinen et al., 2010). In FOCI the modeled transport is too weak by 3-5 Sv compared to the observed (32.1 Sv, Meinen et al., 2010), depending whether the transport is directly estimated in the narrow strait between Florida and the Bahamas or further upstream between Florida and Cuba. Such a 10% error is a common feature of

coarse-resolution Ocean GCMs, since at 1/2° resolution such narrow passage is represented by only three velocity grid cells, even considering that free slip sidewall boundary conditions are used.

In contrast, the transport through the Indonesian Archipelago, representing the exchange between the Pacific and Indian Ocean, is at the upper end of observational estimates with around 15.2 Sv (13 Sv, Gordon et al., 2010). Similarly for the 24 Sv transport through the Mozambique Channel (16.5 Sv, Ullgren et al., 2012). Both transports represent a large portion of the upper

limb of the global overturning circulation, even though they are also determined by the regional and basin-scale wind-driven circulations. After initializing the model from rest, both transports stabilize quickly (Fig. 11). In contrast, transport through Drake Passage is subject to a stronger and prolonged decline over the first 300 model years, starting at 160 Sv and leveling off at around 83 Sv. Hence, ACC strength is rather weak compared to present day estimates of 134 Sv (Cunningham et al., 2003) and 173 Sv (Donohue et al., 2016), respectively. Thereby, FOCI-piCtl is in line with the lower end of historical model

simulations in CMIP3 (144.6 ± 74.1 Sv, Sen Gupta et al., 2009) and CMIP5 (155 ± 51 Sv, Meijers et al., 2012), respectively. Note, that the comparison of volume transports between FOCI-piCtl and present day conditions is limited.

The Atlantic Meridional Overturning stream function (AMOC) is certainly the most important metric for the ocean circulation. It directly influences the meridional transports of heat, freshwater and anthropogenic $CO_2$, thus regulates climate (not only) in the Atlantic sector. The AMOC is a very delicate quantity, since it encloses a wide range of regimes in the horizontal



**Figure 11.** Transport time series (in Sv) from the complete 1500 years reference FOCI-piCtl simulation, across various sections: (a) Straits of Florida, between Florida - Bahamas (FS-BH, blue) and Florida - Cuba (Cu-FS, red), (b) Indonesian Archipelago, (c) Mozambique Channel and (d) Drake Passage. Sections highlighted in Fig. 10. Mean and interannual standard deviation for each section are given next to each time series. Light blue are annual mean, dark blue 10-yr mean values.

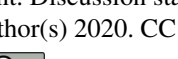



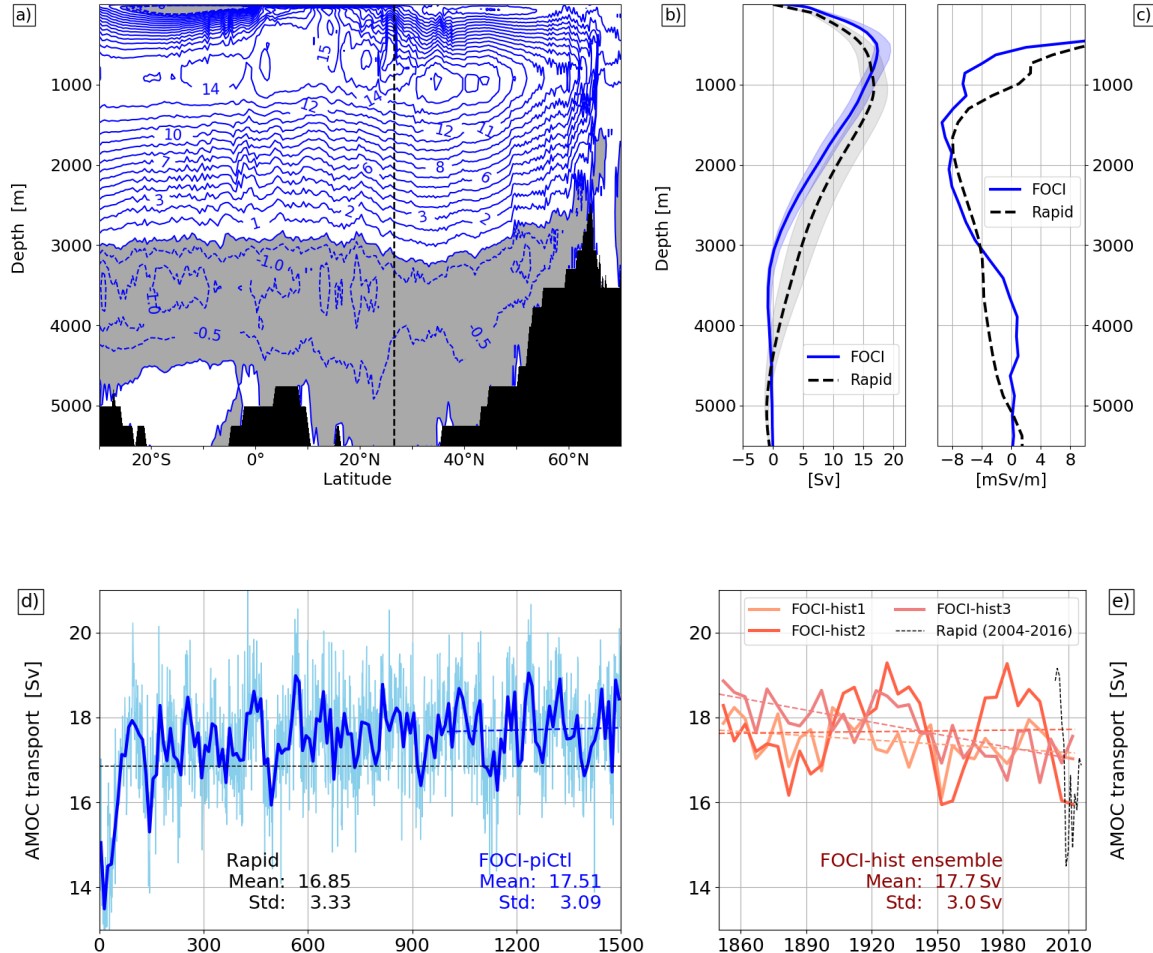

**Figure 12.** Atlantic meridional overturning stream function (in Sv, 500-year mean) as a (a) latitude-depth structure (Shaded in gray are anticlockwise (C.I. = 0.5 Sv), in white are clockwise (C.I. = 1 Sv) cells) and (b) vertical profile at 26.5°N. Provided in dashed black are observational values from the RAPID array, shaded envelopes represent the 5% and 95% quantiles of the annual profile time series. (c) Profile of the z-derivative of AMOC at 26.5°N (in mSv/m). d) Time series of maximum AMOC transport at 26.5°N in the upper 1500 m (in Sv) as 10 year mean (blue) and annual mean (light blue) from FOCI-piCtl, black dashed line represents the mean over the observed 2004-2016 RAPID data; mean and standard deviations are based on monthly means, given at bottom of figure d) for RAPID (black, 2004-2016) and FOCI-piCtl (0-1500 years). e) 5 year mean time series from historical FOCI simulations (FOCI-hist[1,2,3], all orange), linear trends over the whole time period computed from annual means (orange dashed lines) and annual mean RAPID data (black dashed); mean strength and standard deviation calculated from monthly means of the ensemble mean are provided for the period 1850–2013.





and vertical circulation, and combines aspects of wind-driven as well as thermohaline dynamics through its integral character. Small details, e.g. in the formation and transport of dense water masses, can directly reflect on the strength and stability of the AMOC. Under pre-industrial forcing, the AMOC shall not show any long-term trend, but variability on monthly, interannual, multi-decadal and even centennial timescales (Biastoch et al., 2008a; Park and Latif, 2008).

Fig. 12a demonstrates that the overall AMOC structure is well represented in FOCI: a North Atlantic Deep Water (NADW) cell with a maximum of 16.9 Sv and an Antarctic Bottom water Cell (AABW) of up to 1 Sv. The AMOC strength can be evaluated at 26.5°N where observations exist through the RAPID array (McCarthy et al., 2015). Although the absolute transport of the NADW cell fits quite well within half a Sverdrup, maximum transports are shifted towards shallower depth (Fig. 12b). In contrast, the lower water column reflects a too shallow return flow of NADW. While the southward return flow reaches down
at depth of classical upper NADW, the depth of classical lower NADW is not connected to southward flow anymore a typical deficit seen in coarse-resolution OGCM (Biastoch et al., 2008a).

Over the course of the 1500 years pre-industrial control simulation, the AMOC is subject to strong monthly and interannual variability, and compares well to the range of RAPID, with a negligible model-related drift of 0.0018 Sv per decade (Fig. 12e). Two of the three members of the FOCI-hist ensemble show a small but significant ($p$ =0.05) decline in AMOC strength
(-0.03/0.01/-0.09 Sv per decade based on annual mean values (Fig. 12e).

The mean distribution of SSTs shows the typical pattern with higher temperatures in the tropics and lower temperatures in the subpolar to polar regions, with deviations from the zonality only in the region of the WBCs (Fig. 13a). On a global scale, FOCI-piCtl has a cold SST bias (Fig. 13b) which is expected from the comparison of preindustrial conditions with recent observations. However, it also persists for present-day values, although weaker, in the ensemble mean of FOCI-hist (not explicitly
shown). FOCI shows typical SST biases known from all state-of-the-art climate models (Wang et al., 2014). The most prominent SST biases are: a cold anomaly of up to -8 K in the North Atlantic, warm SST biases in the eastern boundary upwelling areas (off Africa, South and North America) of up to +6 K as well as in large parts of the Southern Ocean of up to +3 K. A more detailed discussion of SST biases is found in section 3.4.

The climate system comprises a large range of time scales and hence needs time to adjust to equilibrium. In addition, approx-
imations in the model physics such as parametrizations of unresolved processes and an unbalanced dynamical state during initialization will lead to model drift (Sen Gupta et al., 2012). The temporal evolution of globally averaged temperature and salinity profiles for the pre-industrial control simulation are shown in Fig. 14. As the processes in the upper ocean are faster than it in the deep ocean, upper ocean temperatures equilibrate starting from year 500 on, deeper ocean temperatures tend to drift throughout the 1500-year long integration. This could be related to the imbalance of the radiation at TOA with +0.7 W/m$^2$
(see section 3.1) providing an continuous additional heat into the coupled system, of which the largest part is stored in the ocean (Palmer and McNeall, 2014). In contrast to radiation, the freshwater budget is closed, as salinity in the deep ocean remains unchanged after the initial shock phase, while changes in the upper part are related to freshwater redistribution at the beginning of the integration (Fig. 14b). The described model drift in FOCI is comparable to CMIP3 (Sen Gupta et al., 2012) and CMIP5 (Hobbs et al., 2016) models.





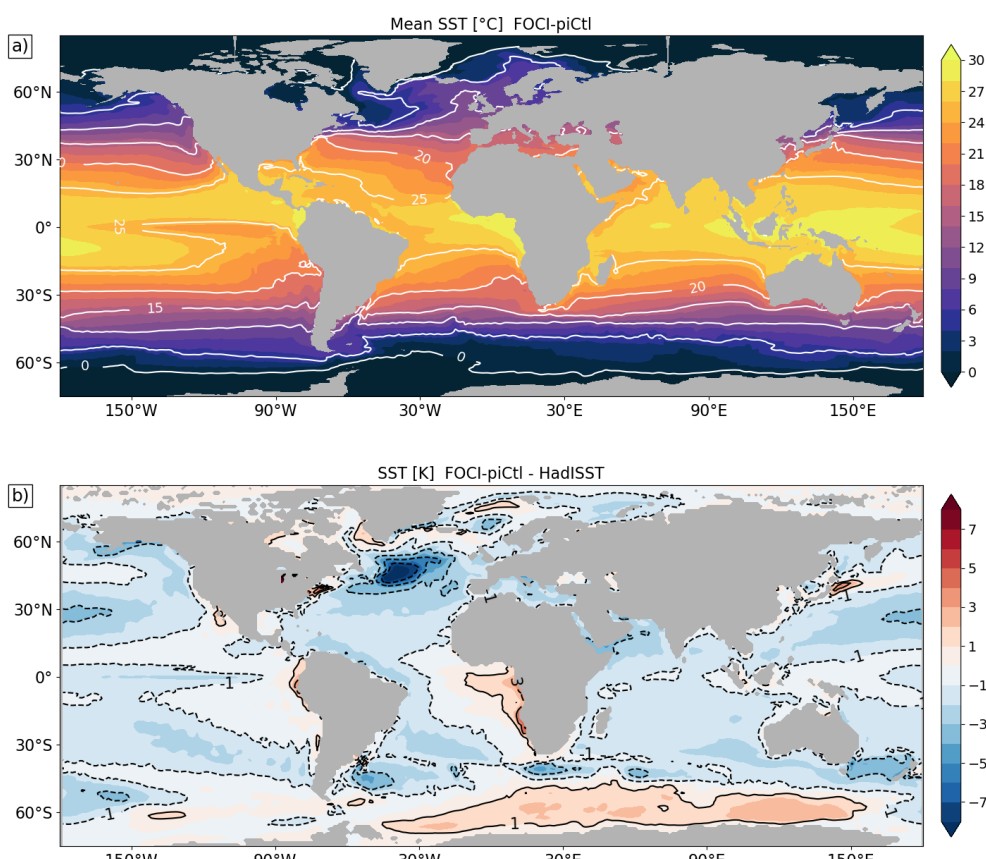

**Figure 13.** (a) Mean SST in °C of the FOCI-piCtl reference simulation averaged over the last 500 years, (b) FOCI-piCtl SST bias (in K) w.r.t observed HadISST (1980-2013).

## 3.3 Sea Ice

Validation of the sea-ice simulation results focuses on the satellite era after 1979 and before the 2000s, in which Arctic sea ice began to decline rapidly in volume and area due to global warming. FOCI simulates a realistic sea-ice cover in the NH but underestimates SH sea-ice coverage and thickness (Fig. 15). The ice-thickness distribution in the Arctic is characterized by the typical thickness increase from the Siberian coast toward Greenland and the Canadian Archipelago, where the ice grows dynamically as thick as 6 m in the model simulations (Fig. 15a). This distribution is an indicator of a realistically simulated ice-drift pattern because both the Transpolar Drift Stream and the anticyclonic Beaufort Gyre promote ice accumulation and export north of and through passages east and west of Greenland. The average modal thickness in the Arctic Ocean for the





**Figure 14.** FOCI-piCtl area-weighted global anomalies w.r.t. the initial state for a) potential temperature in °C) and b) salinity as a depth-time graph. White contour lines indicate 0.4°C temperature and 0.04 salinity intervals above 1000 m, 0.1°C for temperature and 0.01 for salinity below 1000 m, respectively. Also note different vertical scale below 1000 m.

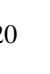



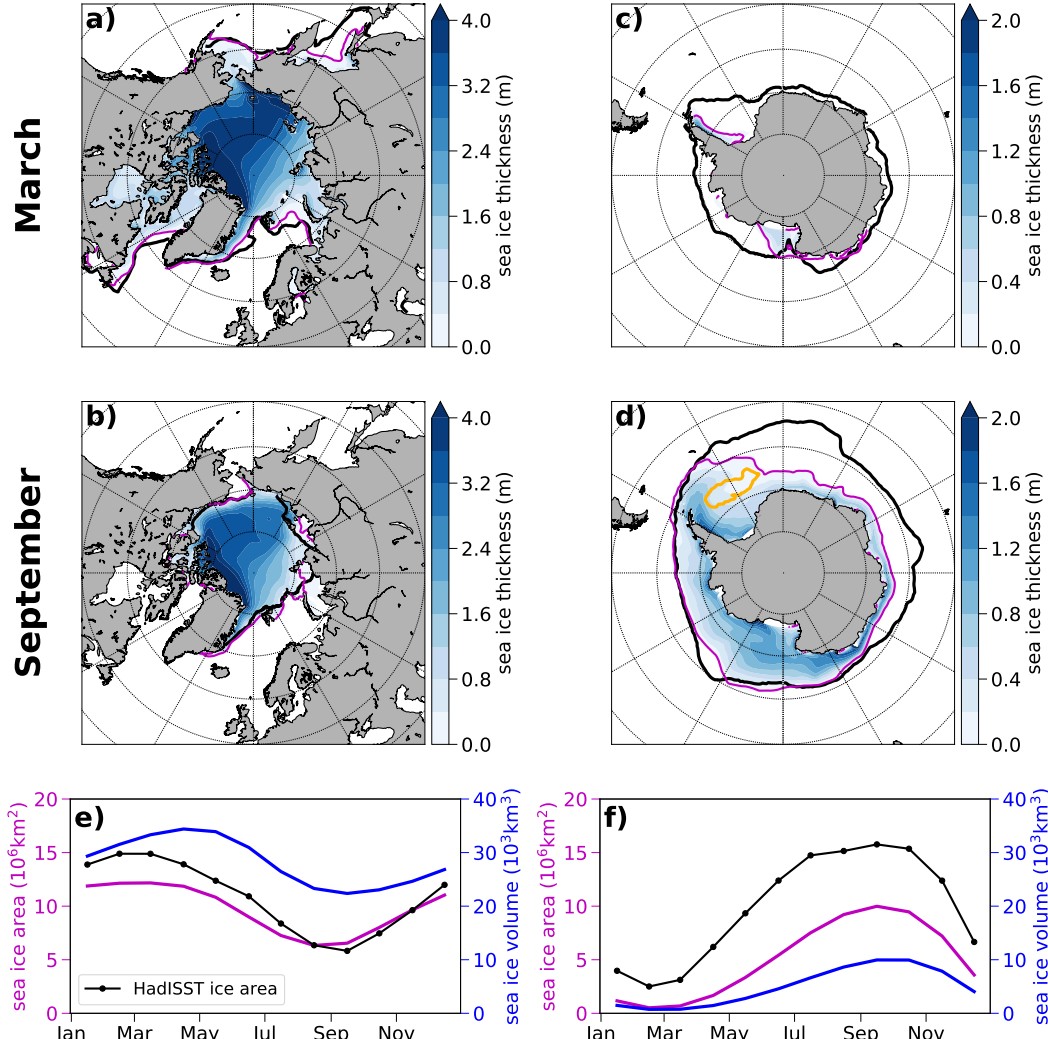

**Figure 15.** Average NH (a,b,e) and SH (c,d,f) sea-ice thickness, ice-edge position and annual cycles of total sea-ice area and volume are shown as 20-year means of a historical simulation (FOCI-hist1, 1980-1999). The ice edge is defined by the 15% sea-ice concentration contour shown as magenta outline in (a)–(d). March and September mean represent winter maximum and summer minimum spatial distributions for the Arctic and Antarctic (note, March is austral summer). In the area marked by a yellow outline in (d) mixed layer depth regularly exceeds 1000 m. Annual cycles of sea-ice area (magenta) and volume (blue) depicted in (e) and (f) are derived from monthly mean mean model output of the years 1980-1999. Observational estimates (HadISST data, 1980-1999) of ice-edge position (a–d) and sea-ice area (e,f) are added as bold black lines.

period 1980-1999 is 3.7 m in winter and decreases to 3.4 m in summer in the depicted historical simulation (FOCI-hist1) (Fig. 15a,b). This compares well with the observed winter-mean sea-ice thickness of the 1980s, 3.64 m (Kwok and Rothrock, 2009). The sea-ice extent is similarly well represented as can be seen by comparing modeled (magenta) and observed sea-ice edges





(black). Ice extent is slightly underestimated in winter, however.

In contrast, FOCI has too little sea ice in the Southern Ocean. Sea-ice extent is clearly underestimated throughout the year. In winter this holds in particular for the Atlantic and Indian sectors (30°W–120°E, Fig. 15d). With modal thicknesses of 0.1 m and 1.0 m in summer and winter, sea ice is also too thin, which may explain the lack of extent. Only small patches in the

Weddell Sea along the Antarctic Peninsula and in the Ross Sea survive summer melt (Fig. 15c). The underestimation of sea-ice thickness and extent are likely related to a pronounced warm bias in the Southern Ocean (Fig. 13c) and the regular recurrence of open ocean deep convection in the Weddell Sea, which is simulated by FOCI on a multi-decadal time scale (not shown). During deep convection heat from mid-depth is brought to the surface forming a polynya, i.e. a hole in the sea-ice cover, where strongly enhanced heat loss to the atmosphere further promotes deep convection (Martin et al., 2013). In panel (d) the potential

polynya region is indicated by a yellow outline. As many coarse-resolution climate models, FOCI with its 1/2° ocean relies on such intense open ocean deep convection to maintain a realistic AMOC bottom cell (Fig. 12a) because dense water formation on the Antarctic shelf is not adequately resolved to form sufficient amounts of bottom water.

Variability of both sea-ice area and volume is dominated by pronounced seasonal cycles presented as climatologies of the period 1980-1999 in Figure 15e,f for the NH and SH. Arctic ice area varies between $6.3 \times 10^6$ and $12.2 \times 10^6$ km$^2$ peaking in March.

While the summer minimum matches observations (HadISST data (Rayner et al., 2003), black curve) well in magnitude, the modeled minimum occurs 1–2 weeks early thus showing the smallest value in August instead of September (magenta line). Arctic sea-ice volume ranges from $22.4 \times 10^3$ km$^3$ in September to $34.4 \times 10^3$ km$^3$ in April. Compared to the often cited PIOMAS ice-ocean hindcast (Schweiger et al., 2011) this is at the upper end of interannual variability but still within two standard deviations of the 1980-2017 mean. In contrast, SH sea-ice area and volume are smallest in late February and largest

in September. With a range of $0.5–10.0 \times 10^6$ km$^2$ the modeled ice area (Fig. 15f, magenta line) is 1.5 (winter) to 5 times (summer) smaller than the observational record (black). Here, we can only speculate about the reasons for the SH discrepancy (see above), but this issue will be central for the next FOCI version.

### 3.4   Ocean Nesting and Impact on Mean State

The unique capability of the new FOCI system is the optional enhancement of ocean resolution in a specific region of interest

within the fully coupled global model which can be optionally coupled to atmospheric chemistry or biogeochemistry in the ocean. The enhanced ocean resolution is achieved by two-way-nesting in the ocean component. So far, two separate nests have been implemented, one over the South Atlantic / western Indian Ocean (INALT10X, see Schwarzkopf et al., 2019, for a description of the ocean-only configuration) and the other one over the North Atlantic (Fig. 2).

Individual experiments are conducted with one of the nest configurations active at a time (Tab. 2). The ocean resolution in

the nested area is enhanced from 1/2° to 1/10° thereby explicitly resolving the mesoscale. The nesting approach allows for centennial climate simulations while resolving mesocale ocean dynamics in specific regions where it is inevitable to capture both the local features, such as the Agulhas or the Gulf Stream current dynamics, and their impact on the large-scale circulation for a most realistic representation of the ocean and its interaction with the atmosphere.

As an example for the significant improvement of the climate mean state in the nested configurations, we show two of the



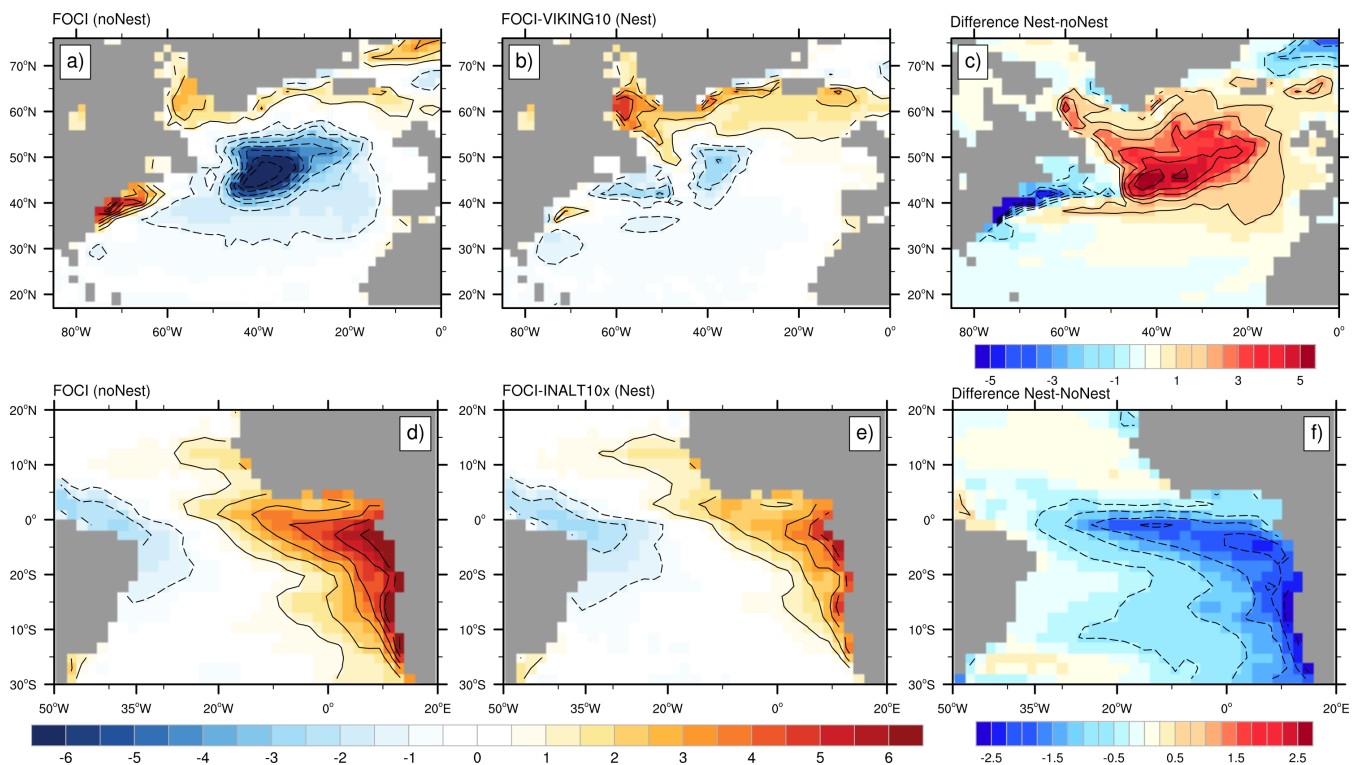

**Figure 16.** FOCI SST anomalies with respect to observed SSTs (HadISST, 1870-1900) in Kelvin: (a) annual mean FOCI unnested and (b) FOCI_VIKING10-piCTL nested configuration in the North Atlantic; boreal summer (July–Sept.) (d) FOCI-piCTL unnested and (e) FOCI_INALT10X-piCTL nested configuration in the tropical Atlantic. Shared color scale for (a-b),(d-e), contour interval of 1 K for (a-e) and 0.5 K for (f). The global mean SST has been removed from FOCI as well as HadISST data prior to anomaly calculations to eliminate the influence of different mean states.

most prominent biases in current climate models (see also Fig. 13) and how they improve by regionally enhanced resolution: the North Atlantic cold bias and the Tropical Atlantic warm bias. Previous studies reported that the first bias can reach -10 K locally (Drews et al., 2015) and average to -4 K over the region 55°W-15°W, 25°N-50°N (Wang et al., 2014, based on CMIP5 models) in the annual mean. In FOCI (with 1/2° ocean grid resolution) this bias amounts to -2.2 K over the same region but

5    SST deviates locally by up to 8 K from reanalysis (Fig. 16a). The cold bias is often associated with the lack of mesoscale dynamics and the path of the separated Gulf Stream in non-eddying ocean models (e.g. Greatbatch et al., 2010). With ocean grid refinement at 1/10°, SSTs in the North Atlantic increase significantly by up to 5 K along the North Atlantic Current path as the simulation of the current's meanders and eddies is significantly improved including variability at the Northwest Corner (Fig. 16b,c). The SST along the North American coast decreases indicating an earlier and more realistic separation of the Gulf

10   Stream from the coast.

The SST warm bias in the Tropical Atlantic has been reported more than a decade ago (Davey et al., 2002) and still persists in current climate models with magnitudes often exceeding +5 K (Richter and Xie, 2008; Richter, 2015; Xu et al., 2014). In



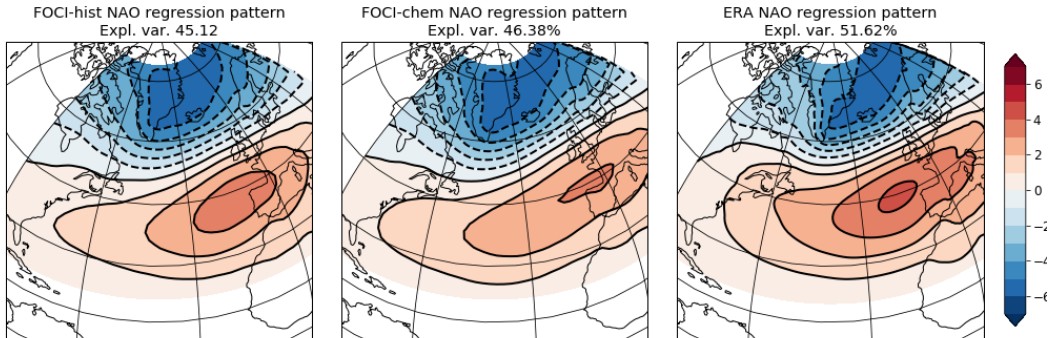

**Figure 17.** Ensemble mean NAO regression pattern from (a) FOCI-hist, (b) FOCI-chem and (c) ERA-Interim for the period 1960 to 2013.

FOCI SSTs are also too high in the eastern equatorial Atlantic basin culminating in the largest deviations of about +6 K along the African coastline (Fig. 16d), similar to the ensemble-mean SST bias in CMIP5 and FOCI's predecessor KCM (Wahl et al., 2009; Harlaß et al., 2018). Increasing ocean resolution from 1/2° to 1/10° reduces the warm bias in FOCI by about 2.5 K in the cold tongue and Benguela upwelling region (Fig. 16e,f). This supports findings by Small et al. (2015), who focused on the Benguela upwelling system. Our results support the notion that an eddy-resolving ocean model together with enhanced resolution in the atmosphere Tozuka et al. (2011); Voldoire et al. (2014); Harlaß et al. (2018) could potentially lead to a more realistic simulations of the equatorial climate. A more detailed analysis of the nested coupled configurations is subject of forthcoming studies.

## 4   Variability

### 4.1   North Atlantic Oscillation

Besides the simulation of a reasonable mean state, a climate model's performance is also evaluated based on its ability to reproduce realistic climate variability. One of the most important and most studied modes of variability is the North Atlantic Oscillation (NAO, Hurrell et al. (2003); Greatbatch (2000)). The NAO is the dominant mode of atmospheric variability in the North Atlantic - European sector, defined by sea level pressure (SLP), often as the first empirical orthogonal function (EOF) in the winter season (December - January - February, DJF). The Icelandic Low and the Azores High strengthen and weaken jointly, determining strength and direction of the westerly winds in the North Atlantic mid-latitudes and thereby surface air temperature and precipitation in Europe and North America.

In the FOCI-hist and FOCI-chem ensembles, the NAO is clearly the leading mode of variability as indicated by the normalized EOF regression patterns derived from detrended SLP in winter (DJF) in the North Atlantic region (20°-80°N, 90°W-40°E,



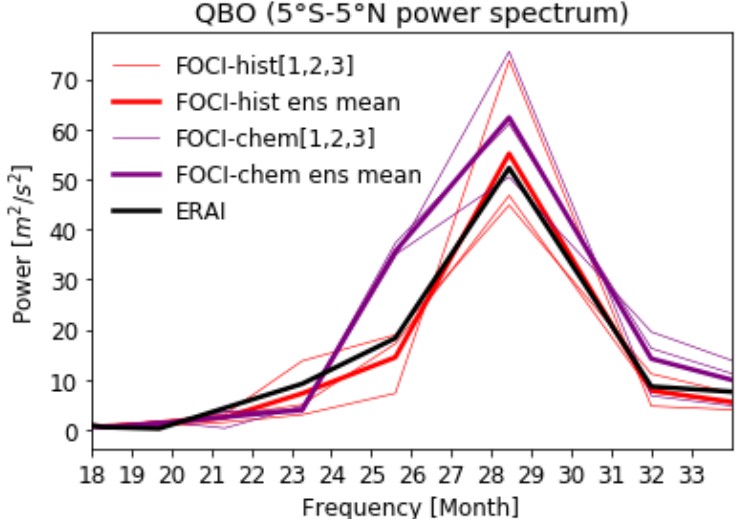

**Figure 18.** QBO spectrum calculated using Welch's method (Welch, 1967) for FOCI-hist (1960–2013; red) and FOCI-chem (1960–2013; purple) simulations as well as ERA-Interim (solid line; 1979–2017) (thin lines are single experiments, thick lines are the ensemble means). For the QBO the zonal mean zonal winds from 5°S to 5°N and 50 hPa (?) are used.

figure 17), and compares well with the pattern derived from a combined ERA40/ERA-Interim reanalysis dataset covering the years 1960 to 2017. The explained variance of the total variance is slightly smaller in both FOCI ensembles (45.1% and 46.4% for FOCI-hist and FOCI-chem respectively), with respect to the reanalysis dataset (51.6%). Compared to reanalysis data, FOCI simulates weaker variability of the Azores High, in FOCI-chem even less than in FOCI-hist. Towards Europe, contour lines in

FOCI are not as close together as in reanalysis data, indicating a slight underestimation of the westerly wind strength in FOCI. On the North American side, the zero lines, denoting the position of the strongest winds, are further to the south in FOCI, over Newfoundland, whereas in ERA, it is located over the Labrador Sea. This might influence the simulation of oceanic deep convection in the Labrador Sea, an important region for deep water formation and very sensitive to small changes in atmospheric variability.

The corresponding time series of the first EOFs are shown in Figure A6. All time series exhibit variability on interannual to decadal time scales with a similarly strong decadal peak in the power spectra of the model simulation (approximately 9.5 years, not shown explicitly) whereas ERA shows a peak at sub-decadal time scales (approximately 8 years, not shown explicitly).

## 4.2   Quasi-Biennial Oscillation

The Quasi-Biennial Oscillation (QBO) dominants the variability in the tropical lower to upper stratosphere (Baldwin et al.,

2001). Downward propagating easterly and westerly zonal mean wind regimes with an average period of 28 months are also seen in temperature and ozone. The QBO is not limited to the tropics but also influences the mean state and variability in the extratropics as well as in the polar stratosphere through changes in the propagation properties for planetary waves and their





interaction with the mean flow. During QBO westerly (QBOw) conditions the polar vortex is cold and more stable, whereas during QBO easterly conditions (QBOe), it is warmer and more disturbed (Holton and Tan, 1980, 1982; Anstey and Shepherd, 2014). Thus, simulating the QBO in a climate model interactively provides a more realistic stratospheric variability and even has an effect on tropospheric variability (Hansen et al., 2016).

One important feature of FOCI is the self-consistently generated QBO (see section 2, a feature that only a few CMIP-type climate models have. The average period of the observed QBO, here represented by the ERA-interim reanalysis, is 28-months (Fig. 18). This is very well captured in both sets of FOCI simulations with and without interactive chemistry, whereas the variability of the individual realizations of both FOCI sets is quite large (from 40 to 70 $m^2/s^2$). The width of the power spectrum is slightly larger in the FOCI-chem experiments as compared to FOCI-hist indicating that the length of the QBO is

affected by the interactive chemistry. In particular the QBO period in FOCI-chem is slightly shorter which will be discussed in a separate paper.

### 4.3   Stratospheric Sudden Warmings

Stratospheric sudden warmings (SSWs) are large disturbances of the NH winter stratospheric polar vortex where the normal circulation breaks down. They are induced by the interaction of upward propagating planetary waves from the troposphere with

the zonal mean flow in the stratosphere (Matsuno, 1971). They were first discovered by Scherhag (1952) and occur every other year (Erlebach et al., 1996; Labitzke and Naujokat, 2000; Charlton and Polvani, 2007) and hence dominate the interannual variability of the stratosphere. The break-down of the vortex is accompanied by a strong temperature increase in the polar stratosphere and a reversal of the normal westerly winds at 60°N and 10 hPa. This is called a major SSW, according to the definition of the World Meteorological Organization (WMO, Labitzke and Naujokat, 2000). SSWs are a prominent example

of stratosphere-troposphere coupling. Their influence can descend to the troposphere and affect surface weather and climate (Quiroz, 1977; Baldwin and Dunkerton, 2001; Thompson et al., 2002; Mitchell et al., 2013; Karpechko, 2017) as well as the ocean circulation (Reichler et al., 2012; Haase et al., 2018). Therefore a good representation of SSWs in climate models is a pre-requisite to improve the tropospheric prediction skill (Baldwin and Dunkerton, 2001; Thompson et al., 2002; Mitchell et al., 2013) and a good test whether and how stratosphere-troposphere coupling is represented in the model.

Figure 19 shows the seasonal distribution (November through April) of major SSWs for ERA and the individual FOCI simulations with and without chemistry in relative frequency of warming events per month and year. The observed SSW distribution (ERA (ERA-Interim), grey (black) bars) is well known: the frequency of SSWs increases steadily from November until it maximizes in January and decreases steadily afterward until the end of the winter. The average frequency of SSWs for the period 1958-2017 (1979–2017) obtained for the combined ERA/ERA-Interim (ERA-Interim period) is 0.52 (0.44) events per year.

Both FOCI versions with (purple bars) and without (red bars) stratospheric chemistry capture the overall SSW total frequency very well with an average of 0.44 and 0.37 warmings. However, a significant deviation is visible for the seasonal distribution: FOCI has the tendency to generate very early SSWs already in November, a time where in observations almost no SSW has ever been observed. This is a known problem of the atmospheric model ECHAM (e.g.  Charlton et al., 2007). It also underestimates significantly the occurrence of SSWs in December and January and shows the maximal occurrence in February. There is also

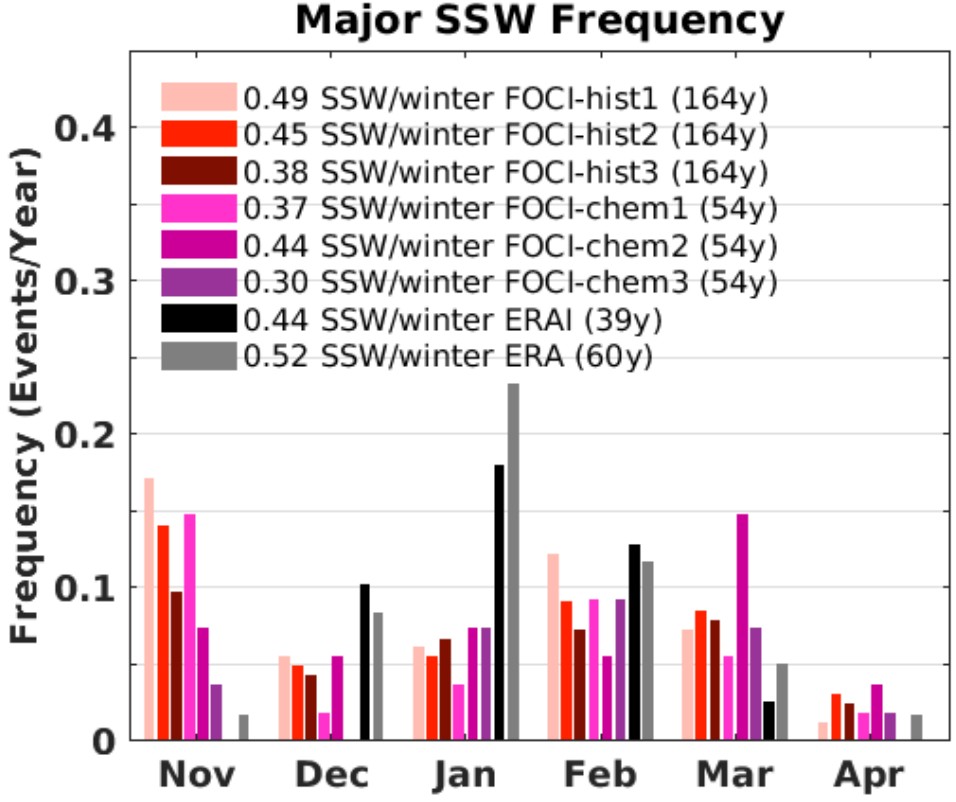

**Figure 19.** The frequency of major SSWs from the FOCI-hist (1850-2013; red) and FOCI-chem (1960-2013; purple) simulations as well as for ERA-Interim (1979–2017; black) and a combined ERA40/ERA-Interim dataset (1958–2017; grey) during NH winter from November through April. The total SSW frequency for the entire winter season is given in the figure.

a tendency of too much variability in March when the polar vortex breaks down for summer conditions. The exact simulation of the seasonal distribution of SSWs is a common problem of climate models (e.g. Charlton et al., 2007; Shaw et al., 2014).

### 4.4 Arctic sea-ice decline

The decline of Arctic sea-ice area due to anthropogenic warming since the 1980s is well simulated by FOCI. Figure 20 shows
5  NH September-mean sea-ice area for the historical period. The modeled ice area evolves stably with realistic interannual variability until the 1980s, when the retreat begins. From 1990 onward the ensemble-mean ice area of the historical simulations (red and purple lines) clearly separate from the pre-industrial control run (blue). With $-0.40 \times 10^6$ km$^2$ per decade (1979–2013) the decline is slightly stronger for the ensemble with stratospheric chemistry (purple dashed line) compared to the one without. The chemistry ensemble also has greater variability among its members, which is indicated by the shaded area
10  depicting two standard deviations computed from the small ensemble. Notably, the observed ice area (black) is well within

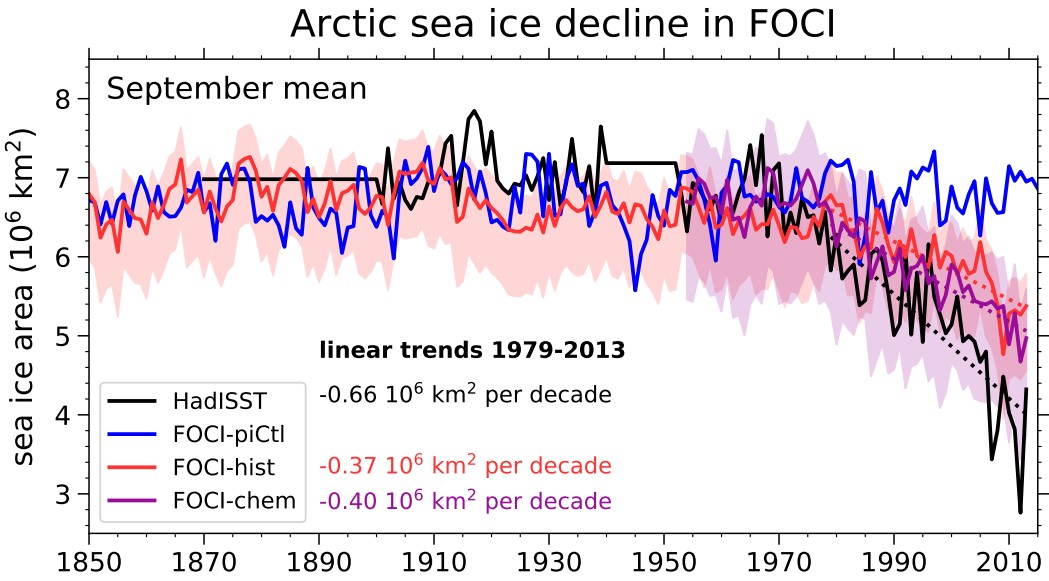

**Figure 20.** Total NH sea-ice area from pre-industrial control (blue) and historical simulations without (red) and with stratospheric chemistry (purple). For the latter mean (bold lines) and range of two standard deviations (shading) are depicted for each small ensemble. The bold black line depicts the HadISST observational estimate for reference. Linear trends over the period 1979–2013 are provided next to the legend and indicated by dashed lines in respective colors.

this uncertainty envelope except for the most extreme years 2007 and 2012. Also, a run with chemistry features the strongest individual trend of $-0.46 \times 10^6$ km$^2$ per decade among all simulations. All trends are statistically significant ($p = 0.01$). Although FOCI underestimates the observed trend by about one-third (from HadISST, black curve in Fig. 20) it performs better than most CMIP5 historical simulations and the ensembles presented here have similar (hist, red) or stronger trends (chem, purple) than

the multi-model mean (Stroeve et al., 2012; Shu et al., 2015).

## 4.5 El Niño/Southern Oscillation

El Niño/Southern Oscillation (ENSO) is the most dominant mode of climate variability on interannual time scales and has influence on extreme weather in the tropical Pacific and beyond (Philander, 1990). ENSO is generated by the coupling of the oceanic thermocline slope with the atmospheric Walker Circulation via the surface wind-SST feedback (Neelin et al., 1998). Its

warm phase, El Niño, is associated with above normal SSTs in the central and eastern equatorial Pacific, weaker trade winds and a relaxed thermocline slope (the cold phase, La Niña, vice versa, Philander, 1990). The overall amplitude of ENSO in FOCI, here represented by the standard deviation of SSTs in the Niño3.4 region (170°W-120°W, 5°S-5°N), is quite realistic, but the seasonal variation is too weak (Fig. 21a). The variability is only slightly increased at the end of the year and too high during boreal summer (Fig. 21a), which indicates that FOCI underestimates the phase locking of ENSO events to the annual

cycle. In terms of periodicity FOCI simulates quite realistic ENSO variability on a frequency band of 3-8 years, but underesti-





mates the short-term variability with a frequency band of 1-3 years (Fig. 21b), which indicates that the recharging-discharging of the heat content may work not efficient enough (Lu et al., 2018). The simulated ENSO is overall quite similar in the different FOCI experiments (Tab. A1). One difference between the piCtl, hist and chem experiments is the increased ENSO variability in the latter two (Fig. 21a,b). This may be explained by the introduction of external forcings in the historical experiments.

Warmer global mean temperatures caused enhanced ENSO variability in the precedessor model, the KCM (Latif et al., 2015), and in addition volcanic eruptions can trigger El Niño events (Khodri et al., 2017).

An important feature of ENSO is its non-linearity and spatial asymmetry (Dommenget et al., 2013; Timmermann et al., 2018): El Niño events are on average stronger than La Niña events and the SST anomalies during El Niño are more in the eastern equatorial Pacific (Fig. 22a), while during La Niña they are more in the central Pacific (Fig. 22b), resulting in an zonal asymmetry,

as shown in Fig. 22c. In FOCI (Fig. 22d-e) El Niño has too much amplitude in the western equatorial Pacific and too little in the eastern equatorial Pacific and La Niña vice versa, resulting in a too weak ENSO asymmetry (Fig. 22f). The underestimation of the non-linearity of ENSO is a common problem in climate models and its potential reasons are still under discussion (Bellenger et al., 2014; Timmermann et al., 2018). One important contribution is the equatorial cold SST bias (Fig. 13), that shifts the climate model into a La Niña-like mean state with too little convection and precipitation over the western equatorial

Pacific (Fig. A7b,c), which in turn hampers the wind-SST feedback (Kim et al., 2014; Wengel et al., 2018; Bayr et al., 2018) and simulated ENSO dynamics (Bayr et al., 2019). Indeed, FOCI has an equatorial cold SST bias (Fig. A7a) and the wind-SST feedback is underestimated by roughly a third (Tab. A1). In summary, FOCI simulates a quite realistic SST variability in the tropical Pacific in terms of amplitude and frequency, but underestimates the non-linearity and phase locking of ENSO similar to most state-of-the-art climate models.

## 5   Conclusions

In this paper we introduced the first version of the Flexible Ocean and Climate Infrastructure (FOCI), a state-of-the-art coupled climate modeling system with the unique ability to explicitly simulate mesoscale ocean eddies in specific regions (examples are given for the Agulhas Current and the Gulf Stream systems) in combination with the stratospheric circulation and chemistry.

FOCI allows to bridge the gap between coarse resolution climate models and global high-resolution weather prediction and ocean-only models to study the evolution of the climate system on regional and seasonal to (multi-) decadal scales. A particular interest is in deciphering internal and external processes driving past, present and future ocean circulation and its role in climate, a topic of utmost socio-economic importance with respect to global warming and international temperature targets.

The first version of FOCI consists of a global high-top atmosphere (ECHAM6.3, T63L95) with a horizontal resolution of 1.8°,

a global ocean model (NEMO3.6, ORCA05) with a horizontal resolution of 0.5°, sea ice (LIM2) and land surface model components (JSBACH), which are coupled through the OASIS3-MCT software package. Depending on the scientific question of interest, optional modules such as stratospheric chemistry (ECHAM6-HAMMOZ) and/or regionally refined nests in the ocean (NEMO-AGRIF) and/or biogeochemistry in the ocean (MOPS) can be added. FOCI thus provides an important step forward



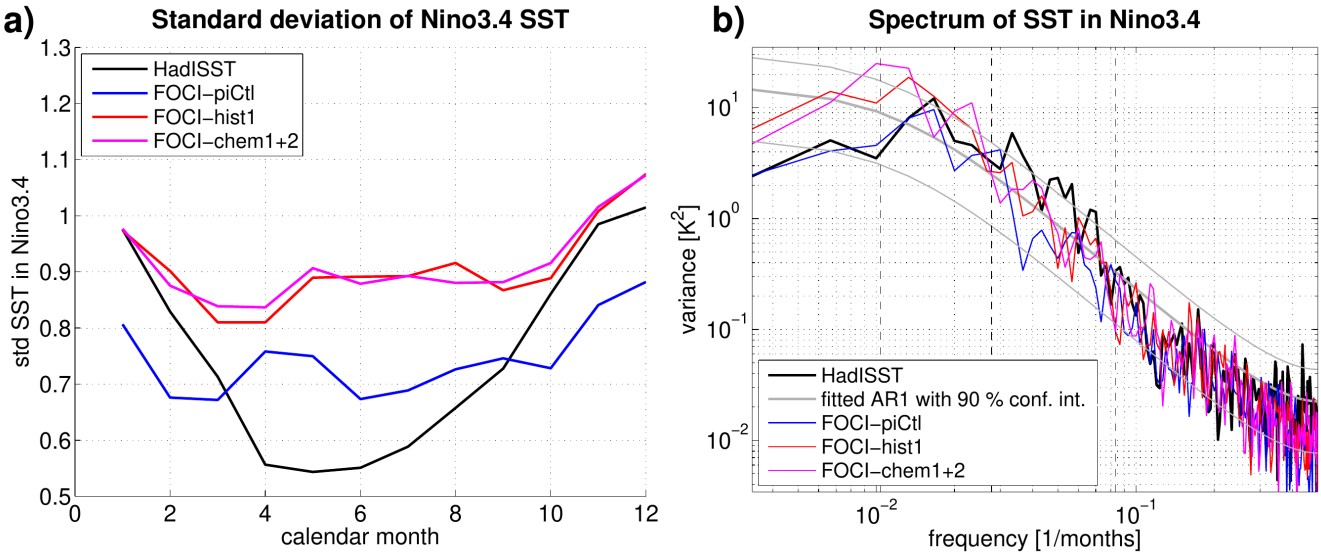

**Figure 21.** For HadISST for the period 1918-2017 and individual FOCI experiments for the last 100 years in a) standard deviation of Niño3.4 SST index for each calendar month; in b) spectrum of SST variability in the Niño3.4 region compared to a fitted AR1 process, the dashed vertical lines mark the 1, 3 and 8 yr frequency, respectively; as the FOCI-chem experiments are only 54 years long, the last 50 years of FOCI-chem1 and FOCI-chem2 are concatenated for this analysis.

to include the atmospheric feedback in multi-decadal regional eddy-rich ocean configurations.

We presented both the scientific vision as well as some technical details and a thorough validation of the different FOCI model components. We assessed FOCI's ability in a 1500-years long pre-industrial control simulation under 1850 conditions in its standard configuration, i.e. T63L95 atmosphere and ORCA05 ocean, three historical simulations without stratospheric chem-
istry from 1850 through 2013, three historical simulations with stratospheric chemistry from 1950 through 2013, as well as two 150-year long coupled nested North and South Atlantic simulations under pre-industrial control simulations.

In the following we summarize the performance of our new model system FOCI in terms of mean state and variability:

- FOCI in its standard configuration runs stably over more than 1500 years and produces a reasonable mean climate
in terms of atmospheric and oceanic temperatures and circulation, as well as sea ice and radiation budget (TOA) in comparison to other coupled climate models of similar resolution.

- The standard FOCI configuration shows a prominent cold bias in SST and SAT in the North Atlantic as well as a pronounced warm bias in the Southern Ocean.

- The climatological vertical structure of zonal mean zonal wind and temperature in the atmosphere compares well with
reanalyses, the variability at polar latitudes is slightly under- and in the tropics overestimated.

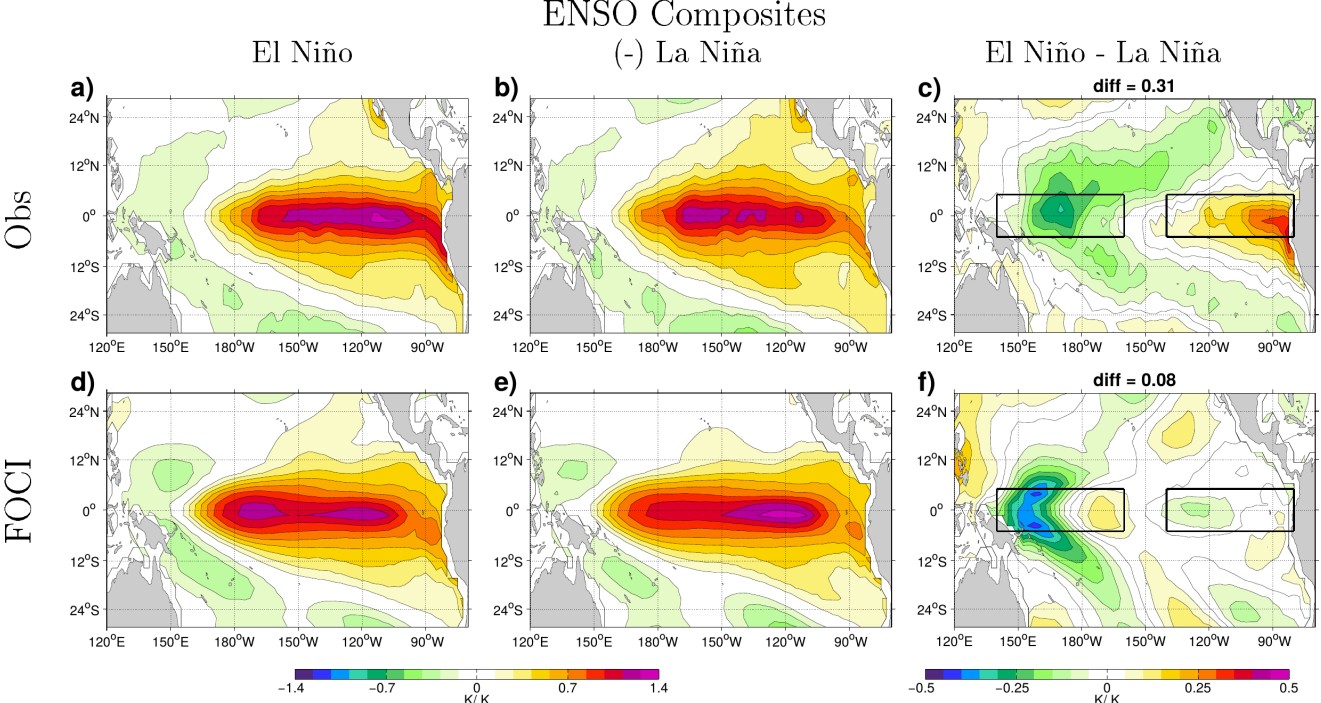

**Figure 22.** ENSO composites of SST anomalies in a-c) for HadISST for the period 1870-2017 and in d-f) for the last 100 years of FOCI-piCtl, with Niño3.4 index as selection criterion and normalized by mean Niño3.4 index; in a) and d) for El Niño (SST in Niño3.4 greater than one standard deviation), b) and e) for La Niña (SST in the Niño3.4 smaller than minus one standard deviation), c) and f) the difference El Niño minus La Niña; The difference of eastern minus western box as shown in c) and f) is given in the header and is a measure for the spatial asymmetry between El Niño and La Niña.

– FOCI with stratospheric chemistry (ECHAM6-HAMMOZ) is able to realistically simulate the evolution of the ozone hole with a maximum in September and/or October and can be used to explicitly resolve the feedback between stratospheric chemistry and atmospheric dynamics. A more detailed study of the effects of stratospheric chemistry on atmospheric dynamics is subject of a future publication.

– The horizontal and meridional cells of the ocean circulation are comparable to observations. Remarkable is the stable AMOC, with a strength comparable to long-term observations (RAPID).

– While FOCI produces a realistic simulation of NH sea-ice distribution and evolution, sea ice on the SH is significantly underestimated. The latter bias is subject to ongoing work and likely related to mixing in the upper ocean.

– The ensemble set of historical simulations of the standard FOCI configuration agrees well with observations and re-
analyses products and captures the tropospheric warming, the stratospheric cooling, and an Arctic sea-ice decline with increasing anthropogenic forcing.





- The simulated AMOC strength compares well with observations in both mean state and monthly to interannual variability. The inconsistency of (multi-)decadal variability among ensemble members hampers a conclusion regarding a possible AMOC decline over the historical period 1850–2013. Nevertheless, two of three ensemble members indicate a significant though weak decline.

- Both nested FOCI configurations which resolve the mesoscale in the ocean reduce the long-standing SST biases in climate models from -8K towards -2.5K in the North and from +6K to +3K in the South Atlantic due to improved representations of the pathway of the North Atlantic Current and improved upwelling in the Benguela region of the South Atlantic (Figure 16). A more detailed study of the effects and consequences of the ocean nests is subject of upcoming papers.

- FOCI is able to reproduce observed climate variability patterns such as ENSO and NAO.

- Even though the NAO is the leading mode of variability in the North Atlantic–European sector in FOCI, westerly winds are weaker in the model. This may influence oceanic deep convection in the Labrador Sea and thus the AMOC.

- FOCI simulates realistic SST variability in the tropical Pacific in terms of amplitude and frequency, but as most state-of-the-art climate models, it underestimates the non-linearity and phase locking of ENSO.

- FOCI is able to realistically simulate the variability of zonal mean zonal winds in the tropical stratosphere with a 28-month QBO, a feature not common to most state-of-the-art climate models.

- FOCI represents the total frequency of stratospheric sudden warmings well, but lacks as most state-of-the-art climate models a realistic seasonal distribution with a peak occurrence of SSWs in January and February.

We demonstrate important progress with FOCI in comparison to other climate models. This includes 1) the significant SST
bias reduction in the North and South Atlantic and 2) the possibility to combine the nested ocean configurations with a coupled atmosphere which includes a realistic high-top atmospheric circulation and chemistry. Nevertheless, we are continuously working on improving our model system. We are currently working on the bias in SH SSTs and the position of the SH westerlies. We are also planning on improving the relatively coarse atmospheric resolution by enhancing the horizontal resolution in the atmosphere to T127 and shifting the calculation of the exchange fluxes between atmosphere and ocean onto the finer resolved
ocean grid. We are currently investigating the ability of the atmospheric model to receive signals from mesoscale ocean eddies and fronts. Another working area is the closure of the carbon cycle which involves an active coupling of the atmospheric chemistry with the land and the biogeochemistry in the ocean. Further, process oriented studies drive the development of other ocean nests such as in the Indian Ocean, the Southern Ocean, and a combination of the North and South Atlantic nests. In the frame of the Helmholtz Earth System Modeling (ESM) initiative, we are currently also striving for a homogenization of the
software structure with the ESM-Tools[5] and are also testing alternatives for the atmospheric model component ECHAM which does not scale very well on high-performance computers due to its spectral dynamical core.

---

[5](https://www.esm-tools.net)



Besides these caveats and further model developments, the new model system allows us to tackle a couple of urgent scientific questions in order to improve the understanding of the ocean's role in climate change. We plan to advance 1) the fundamental understanding of the ocean system, 2) the prediction and attribution of changes in the ocean and in the climate system, and 3) the understanding of the physical drivers of ecosystems and biogeochemical cycles. Therefor, FOCI is the ideal model system,

capable to resolve mesoscale ocean eddies, stratosphere-troposphere-ocean interactions, as well as ocean-biogeochemistry, and allows to run multi-decadal ensemble simulations with reasonable computational resources.

*Code availability.* FOCI is composed of several component models which do not allow us to distribute the full source code due to licensing issues. The full source code for ECHAM6 is available from MPI Hamburg at http://www.mpimet.mpg.de/en/science/models/mpi-esm/echam.html after signing a licence with MPI. The FOCI version presented here is based on ECHAM6 version 6.3.04). The full NEMO

source code is available at https://forge.ipsl.jussieu.fr/nemo/svn/NEMO/releases/release-3.6/NEMOGCM. The revision used in all FOCI versions is revision 6721. All modifications we made to the original ECHAM6 and NEMO source code together with the full runtime environment including namelists settings are provided at http://doi.org/10.5281/zenodo.3568061. All codes to reproduce the figures presented in this manuscript are available from the same location.

*Data availability.* Output data necessary to reproduce the results presented in this publication are available at the same locations as described

in the code availability section.

*Author contributions.* KM and AB initiated, coordinated and supervised the FOCI development and the editing of the paper. SW, JH and TM designed and carried out the experiments with contributions from all other co-authors. All co-authors prepared the manuscript jointly.

*Competing interests.* The authors declare that they have no conflict of interest.

*Acknowledgements.* We are grateful for long-term modeling support of the GEOMAR Helmholtz Center for Ocean Research in Kiel and

numerous discussions with a number of colleagues in Kiel, Hamburg, Grenoble, Toulouse and elsewhere. We are particularly grateful for the continued computing support and resources provided by the North-German Supercomputing Alliance (HLRN) where all simulations have been performed.

# 1  Appendix

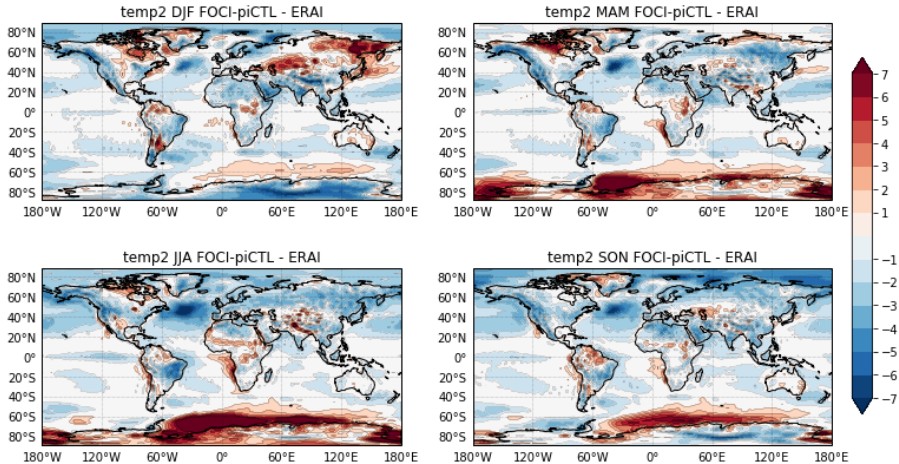

**Figure A1.** Seasonal mean SAT bias (DJF, MAM, JJA and SON from top left to lower right) with respect to ERA-Interim (1979-2017).

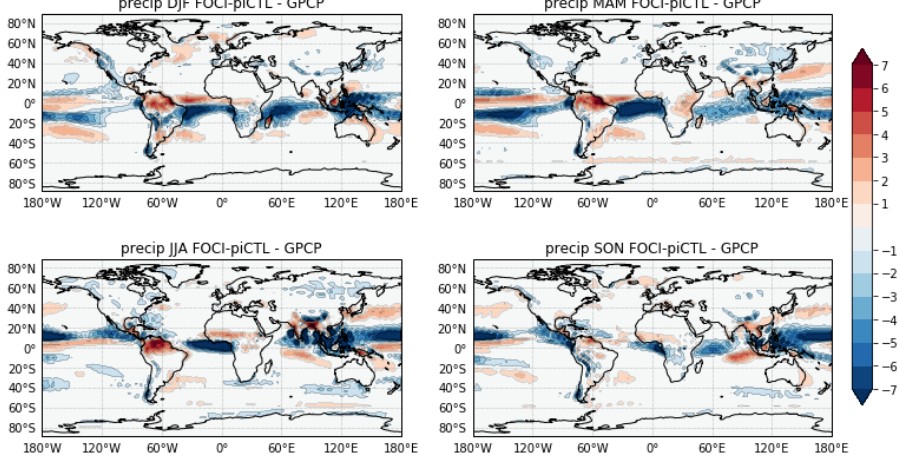

**Figure A2.** Seasonal mean precipitation bias (DJF, MAM, JJA and SON from top left to lower right) with respect to the GPCP dataset (1979-2013).



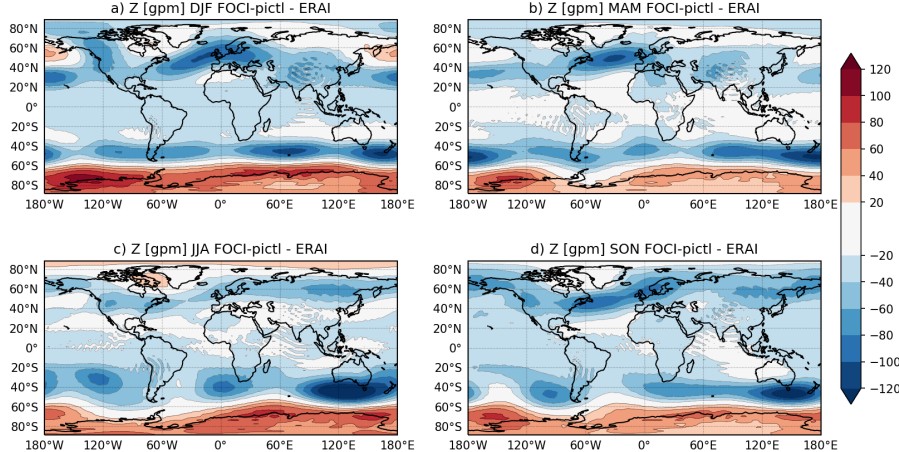

**Figure A3.** Seasonal mean Geopotential bias (DJF, MAM, JJA and SON from top left to lower right) at 500hPa with respect to ERA-Interim (1980-2013).

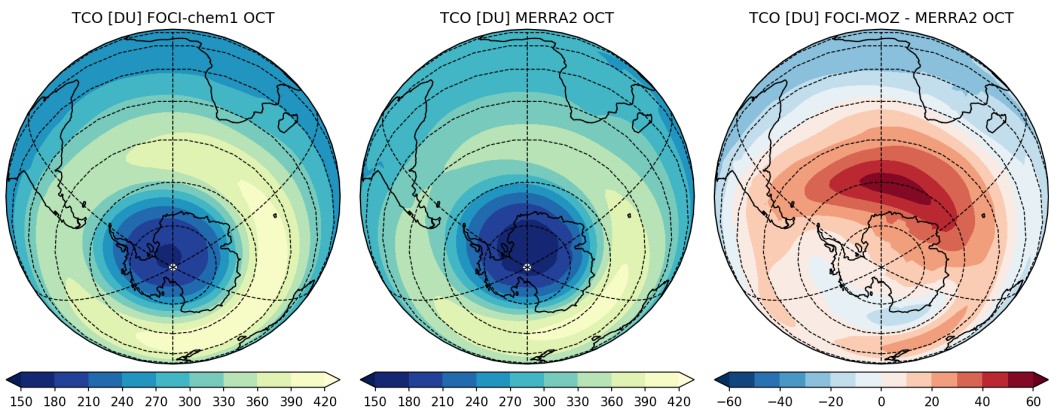

**Figure A4.** Mean TCO distribution in the SH in October in (a) the FOCI-chem1 simulation, (b) the MERRA2 reanalysis, and (c) the differences between (a) and (b) from 1980 to 2013.



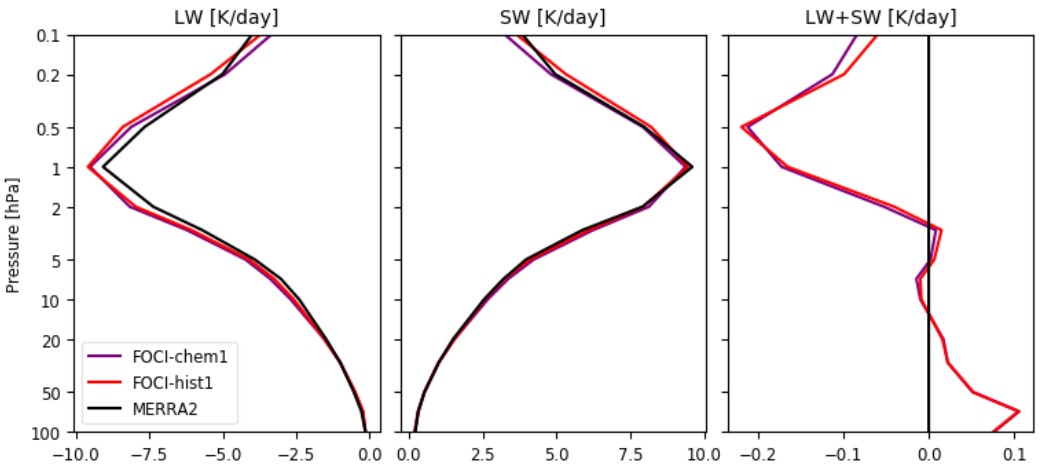

**Figure A5.** Global average vertical profile of SW (left), LW (center) and net (SW+LW) heating rates in FOCI-hist1, FOCI-chem1 and MERRA2 reanalysis data from 1980 to 2013.

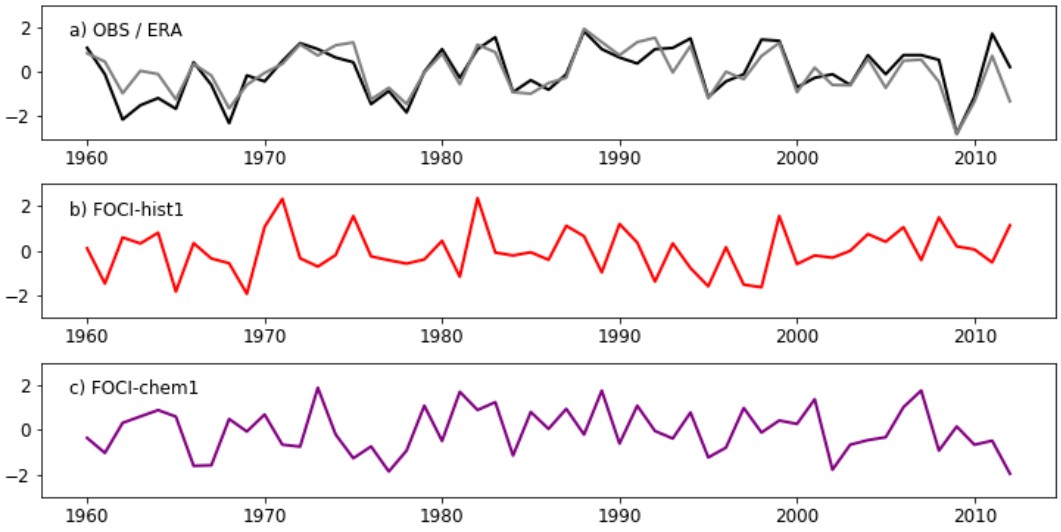

**Figure A6.** NAO timeseries derived from a) Hurrel's station based index (black) and ERA-Interim reanalysis (gray), b) FOCI-hist1 and c) FOCI-chem1. Please see text on how the NAO timeseries are constructed.



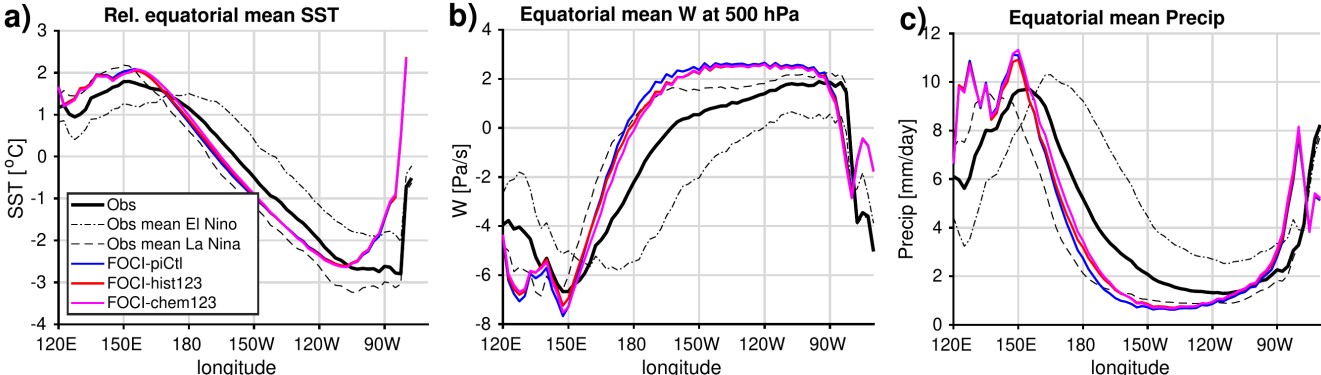

**Figure A7.** Mean state along the equator ($5^o S - 5^o N$) for observations/reanalysis data and FOCI experiments, in a) for relative SST (i.e. area mean of tropical Pacific is subtracted), in b) for vertical wind at 500 hPa and in c) for precipitation.

| | HadISST | FOCI-piCtl | FOCI-hist1 | FOCI-hist2 | FOCI-hist3 | FOCI-chem1 | FOCI-chem2 | FOCI-chem3 |
|---|---|---|---|---|---|---|---|---|
| std Niño3.4 $[K]$ | 0.77 | 0.77 | 0.94 | 0.80 | 0.99 | 1.03 | 0.85 | 0.79 |
| PLI Niño3.4 | 1.71 | 1.09 | 1.14 | 1.20 | 1.12 | 1.13 | 1.09 | 1.07 |
| frq 1-3 yr Niño3.4 $[K^2]$ | 29.1 | 13.2 | 17.7 | 13.1 | 24.1 | 11.0 | 7.2 | 5.4 |
| frq 3-8 yr Niño3.4 $[K^2]$ | 33.2 | 29.6 | 52.5 | 37.5 | 102.9 | 49.7 | 41.1 | 22.2 |
| frq ratio Niño3.4 | 1.14 | 2.24 | 2.96 | 2.86 | 4.26 | 4.53 | 5.74 | 4.11 |
| spatial asymmetry $[K]$ | 0.38 | 0.11 | 0.22 | 0.14 | 0.22 | 0.11 | -0.02 | 0.13 |
| SST bias Niño4 $[K]$ | 0.00 | -0.48 | -0.48 | -0.37 | -0.45 | -0.44 | -0.29 | -0.35 |
| U10 feedback $[m/s/K]$ | 1.56 | 1.08 | 1.22 | 1.27 | 1.21 | 1.22 | 1.23 | 1.31 |

**Table A1.** ENSO properties in Observations (HadISST (Rayner et al., 2003), ERA-Interim (Dee et al., 2011)) and the individual FOCI experiments. In the 1. row the standard deviation (std) of Niño3.4 SST; in the 2. row the Phase Locking Index (PLI) of Niño3.4 SST as defined in Bellenger et al. (2014) as the std of Niño3.4 of December, January and February devided by std of Niño3.4 of April, May and June; in the 3. row the variability of the Niño3.4 SST on a frequency band of 1-3 years; in the 4. row the variability of the Niño3.4 SST on a frequency band of 3-8 years; in the 5. row the ratio of row 4 and row 3; in the 6. row the spatial asymmetry between El Niño and La Niña as shown in Fig. 22c,f; in the 7. row the SST bias in the Niño4 region relative to HadISST; in the 8. row the zonal wind-SST feedback in Niño4, calculated by regression of U10 anomalies of Niño4 onto SST anomalies of Niño3.4.



## Appendix: References

Adler, R. F., Sapiano, M. R. P., Huffman, G. J., Wang, J.-J., Gu, G., Bolvin, D., Chiu, L., Schneider, U., Becker, A., Nelkin, E., Xie, P., Ferraro, R., and Shin, D.-B.: The Global Precipitation Climatology Project (GPCP) Monthly Analysis (New Version 2.3) and a Review of 2017 Global Precipitation, Atmosphere, 9, https://doi.org/10.3390/atmos9040138, 2018.

Anstey, J. A. and Shepherd, T. G.: High-latitude influence of the quasi-biennial oscillation, Quarterly Journal of the Royal Meteorological Society, 140, 1–21, https://doi.org/10.1002/qj.2132, 2014.

Arakawa, A.: Computational design for long-term numerical integration of the equations of fluid motion: Two dimensional incompressible flow. Part I, J. Comput. Phys., 1, 119–143, https://doi.org/10.1016/0021-9991(66)90015-5, 1966.

Arakawa, A. and Hsu, Y.-J. G.: Energy conserving and potential-enstrophy dissipating schemes for the shallow water equations, Mon. Wea.
Rev., 118, 1960–1969, https://doi.org/10.1175/1520-0493, 1990.

Ba, J., Keenlyside, N. S., Park, W., Latif, M., Hawkins, E., and Ding, H.: A mechanism for Atlantic multidecadal variability in the Kiel Climate Model, Climate Dynamics, 41, 2133–2144, https://doi.org/10.1007/s00382-012-1633-4, 2013.

Babington, P.: Scientific Assessment of Ozone Depletion: 2018, Global Ozone Research and Monitoring Project, Geneva, Switzerland, report No. 58, 588p, 2018.

Bacmeister, J. T., Wehner, M. F., Neale, R. B., Gettelman, A., Hannay, C. E., Lauritzen, P. H., Caron, J. M., and Truesdale, J. E.: Exploratory high-resolution climate simulations using the Community Atmosphere Model (CAM), J. Climate, 27, 3073—-3099, https://doi.org/10.1175/JCLI-D-13-00387.1, 2014.

Baldwin, M. P. and Dunkerton, T. J.: Stratospheric harbingers of anomalous weather regimes, Science, 294, 581–584, https://doi.org/10.1126/science.1063315, 2001.

Baldwin, M. P., Gray, L. J., Dunkerton, T. J., Hamilton, K., Haynes, P. H., Randel, W. J., Holton, J. R., Alexander, M. J., Hirota, I., Horinouchi, T., Jones, D. B., Kinnersley, J. S., Marquardt, C., Sato, K., and Takahashi, M.: The quasi-biennial oscillation, Reviews of Geophysics, 39, 179–229, https://doi.org/10.1029/1999RG000073, 2001.

Barnier, B., Madec, G., Penduff, T., Molines, J.-M., Treguier, A.-M., Le Sommer, J., Beckmann, A., Biastoch, A., Böning, C., Dengg, J., et al.: Impact of partial steps and momentum advection schemes in a global ocean circulation model at eddy-permitting resolution, Ocean
dynamics, 56, 543–567, 2006.

Bayr, T., Latif, M., Dommenget, D., Wengel, C., Harlaß, J., and Park, W.: Mean-state dependence of ENSO atmospheric feedbacks in climate models, Climate Dynamics, 50, 3171–3194, https://doi.org/10.1007/s00382-017-3799-2, 2018.

Bayr, T., Wengel, C., Latif, M., Dommenget, D., Lübbecke, J., and Park, W.: Error compensation of ENSO atmospheric feedbacks in climate models and its influence on simulated ENSO dynamics, Climate Dynamics, 53, 155–172, https://doi.org/10.1007/s00382-018-4575-7,
30  2019.

Behrens, E., Biastoch, A., and Böning, C.: Spurious AMOC trends in global ocean sea-ice models related to subarctic freshwater forcing, Ocean Modelling, 69, 39–49, https://doi.org/http://dx.doi.org/10.1016/j.ocemod.2013.05.004, 2013.

Bellenger, H., Guilyardi, E., Leloup, J., Lengaigne, M., and Vialard, J.: ENSO representation in climate models: From CMIP3 to CMIP5, Climate Dynamics, 42, 1999–2018, https://doi.org/10.1007/s00382-013-1783-z, 2014.

Biastoch, A., Böning, C. W., Getzlaff, J., Molines, J.-M., and Madec, G.: Causes of interannual-decadal variability in the meridional overturning circulation of the mid-latitude North Atlantic Ocean, J. Clim., 21, 6599–6615, https://doi.org/10.1175/2008JCLI2404.1, 2008a.





Biastoch, A., Lutjeharms, J. R. E., Böning, C. W., and Scheinert, M.: Mesoscale perturbations control inter-ocean exchange south of Africa, Geophysical Research Letters, 35, L20 602, https://doi.org/10.1029/2008GL035132, 2008b.

Biastoch, A., Böning, C. W., Schwarzkopf, F. U., and Lutjeharms, J. R.: Increase in Agulhas leakage due to poleward shift of Southern Hemisphere westerlies, Nature, 462, 495–498, https://doi.org/10.1038/nature08519, 2009.

Biastoch, A., Durgadoo, J. V., Morrison, A. K., van Sebille, E., Weijer, W., and Griffies, S. M.: Atlantic multi-decadal oscillation covaries with Agulhas leakage, Nature Communications, 6, 10 082, https://doi.org/10.1038/ncomms10082, 2015.

Blanke, B. and Delecluse, P.: Variability of the Tropical Atlantic Ocean Simulated by a General Circulation Model with Two Different Mixed-Layer Physics, J. Phys. Oceanogr., 23, 1363–1388, https://doi.org/10.1175/1520-0485, 1993.

Bodeker, G. E., Shiona, H., and Eskes, H.: Indicators of Antarctic ozone depletion, Atmospheric Chemistry and Physics, 5, 2603–2615,
https://doi.org/10.5194/acp-5-2603-2005, 2005.

Böning, C. W., Behrens, E., Biastoch, A., Getzlaff, K., and Bamber, J. L.: Emerging impact of Greenland meltwater on deepwater formation in the North Atlantic Ocean, Nat. Geosci., 9, 523–527, https://doi.org/10.1038/ngeo2740, 2016.

Brovkin, V., Raddatz, T., Reick, C. H., Claussen, M., and Gayler, V.: Global biogeophysical interactions between forest and climate, Geophys. Res. Lett., 36, https://doi.org/10.1029/2009GL037543, 2009.

Chang, C. Y., Nigam, S., and Carton, J. A.: Origin of the springtime westerly bias in equatorial Atlantic surface winds in the community Atmosphere Model version 3 (CAM3) simulation, Journal of Climate, 21, 4766–4778, https://doi.org/10.1175/2008JCLI2138.1, 2008.

Charlton, A. J. and Polvani, L. M.: A new look at stratospheric sudden warmings. Part I: Climatology and modeling benchmarks, Journal of Climate, 20, 449–469, https://doi.org/10.1175/JCLI3996.1, 2007.

Charlton, A. J., Polvani, L. M., Perlwitz, J., Sassi, F., Manzini, E., Shibata, K., Pawson, S., Nielsen, J. E., and Rind, D.: A new
look at stratospheric sudden warmings. Part II: Evaluation of numerical model simulations, Journal of Climate, 20, 470–488, https://doi.org/10.1175/JCLI3994.1, 2007.

Cunningham, S. A., Alderson, S. G., King, B. A., and Brandon, M. A.: Transport and variability of the Antarctic Circumpolar Current in Drake Passage, J. Geophys. Res., 108, 1–17, https://doi.org/10.1029/2001JC001147, 2003.

Davey, M., Huddleston, M., Sperber, K., Braconnot, P., Bryan, F., Chen, D., Colman, R., Cooper, C., Cubasch, U., Delecluse, P., DeWitt,
D., Fairhead, L., Flato, G., Gordon, S., Hogan, T., Ji, M., Kimoto, M., Kitoh, A., Knutson, T., Latif, M., Treut, H. L., Li, T., Manabe, S., Mechoso, C. R., Power, S., Roeckner, E., Terray, L., Vintzileos, A., Voss, R., Wang, B., Washington, W., Yoshikawa, I., Yu, J., Yukimoto, S., Zebiak, S., and Meehl, G.: STOIC: a study of coupled model climatology and variability in tropical ocean regions, Climate Dynamics, 18, 403–420, https://doi.org/10.1007/s00382-001-0188-6, 2002.

Debreu, L., Vouland, C., and Blayo, E.: AGRIF: Adaptive grid refinement in Fortran, Computers & Geosciences, 34, 8–13,
https://doi.org/10.1016/j.cageo.2007.01.009, 2008.

Dee, D. P., Uppala, S. M., Simmons, A. J., Berrisford, P., Poli, P., Kobayashi, S., Andrae, U., Balmaseda, M. A., Balsamo, G., Bauer, P., Bechtold, P., Beljaars, A. C. M., van de Berg, L., Bidlot, J., Bormann, N., Delsol, C., Dragani, R., Fuentes, M., Geer, A. J., Haimberger, L., Healy, S. B., Hersbach, H., Hólm, E. V., Isaksen, L., Kållberg, P., Köhler, M., Matricardi, M., McNally, A. P., Monge-Sanz, B. M., Morcrette, J.-J., Park, B.-K., Peubey, C., de Rosnay, P., Tavolato, C., Thépaut, J.-N., and Vitart, F.: The ERA-Interim reanalysis:
configuration and performance of the data assimilation system, Quarterly Journal of the Royal Meteorological Society, 137, 553–597, https://doi.org/10.1002/qj.828, 2011.





Delworth, T. L., Rosati, A., Anderson, W., Adcroft, A. J., Balaji, V., Benson, R., Dixon, K., Griffies, S. M., Lee, H.-C., Pacanowski, R. C., Vecchi, G. A., Wittenberg, A. T., Zeng, F., and Zhang, R.: Simulated climate and climate change in the GFDL CM2.5 high–resolution coupled climate model, J. Climate, 25, 2755—-2781, https://doi.org/10.1175/JCLI-D-11-00316.1, 2012.

Ding, H., Greatbatch, R. J., Latif, M., and Park, W.: The impact of sea surface temperature bias on equatorial Atlantic interannual variability in partially coupled model experiments, Geophysical Research Letters, 42, 5540–5546, 2015.

Dommenget, D., Bayr, T., and Frauen, C.: Analysis of the non-linearity in the pattern and time evolution of El Niño southern oscillation, Climate Dynamics, 40, 2825–2847, https://doi.org/10.1007/s00382-012-1475-0, 2013.

Donohue, K. A., Tracey, K. L., Watts, D. R., Chidichimo, M. P., and Chereskin, T. K.: Mean Antarctic Circumpolar Current transport measured in Drake Passage, Geophys. Res. Lett., 43, 11,760–11,767, https://doi.org/10.1002/2016GL070319, 2016.

Drews, A., Greatbatch, R. J., Ding, H., Latif, M., and Park, W.: The use of a flow field correction technique for alleviating the North Atlantic cold bias with application to the Kiel Climate Model, Ocean Dyn., 65, 1079–1093, https://doi.org/10.1007/s10236-015-0853-7, 2015.

Durgadoo, J. V., Loveday, B. R., Reason, C. J. C., Penven, P., and Biastoch, A.: Agulhas Leakage Predominantly Responds to the Southern Hemisphere Westerlies, Journal of Physical Oceanography, 43, 2113–2131, https://doi.org/10.1175/JPO-D-13-047.1, 2013.

Erlebach, P., Langematz, U., and Pawson, S.: Simulations of stratospheric sudden warmings in the Berlin troposphere-stratosphere-mesosphere GCM, Annales Geophysicae, 14, 443–463, https://doi.org/10.1007/s00585-996-0443-6, 1996.

et al., S.: Evidence for a regional warm bias in the Early Cretaceous TEX86 record, Earth and Planetary Science Letters, https://doi.org/submitted, 2019.

Evans, G., McDonagh, E., King, B., Bryden, H., Bakker, D., Brown, P., Schuster, U., Speer, K., and van Heuven, S.: South Atlantic interbasin exchanges of mass, heat, salt and anthropogenic carbon, Progress in Oceanography, 151, 62 – 82, https://doi.org/https://doi.org/10.1016/j.pocean.2016.11.005, 2017.

Fichefet, T. and Morales Maqueda, M. A.: Sensitivity of a global sea ice model to the treatment of ice thermodynamic and dynamics, J. Geophys. Res., 102, 12 609–12 646, https://doi.org/10.1029/97JC00480, 1997.

Frith, S. M., Kramarova, N. A., Stolarski, R. S., McPeters, R. D., Bhartia, P. K., and Labow, G. J.: Recent changes in total column ozone based on the SBUV Version 8.6 Merged Ozone Data Set, Journal of Geophysical Research: Atmospheres, 119, 9735–9751, https://doi.org/10.1002/2014JD021889, 2014.

Gent, P. R. and McWilliams, J. C.: Isopycnal mixing in ocean circulation models, J. Phys. Ocean., 20, 150–155, 1990.

Giorgetta, M. A., Jungclaus, J., Reick, C. H., Legutke, S., Bader, J., Böttinger, M., Brovkin, V., Crueger, T., Esch, M., Fieg, K., Glushak, K., Gayler, V., Haak, H., Hollweg, H.-D., Ilyina, T., Kinne, S., Kornblueh, L., Matei, D., Mauritsen, T., Mikolajewicz, U., Mueller, W., Notz, D., Pithan, F., Raddatz, T., Rast, S., Redler, R., Roeckner, E., Schmidt, H., Schnur, R., Segschneider, J., Six, K. D., Stockhause, M., Timmreck, C., Wegner, J., Widmann, H., Wieners, K.-H., Claussen, M., Marotzke, J., and Stevens, B.: Climate and carbon cycle changes from 1850 to 2100 in MPI-ESM simulations for the Coupled Model Intercomparison Project phase 5, Journal of Advances in Modeling Earth Systems, 5, 572–597, https://doi.org/10.1002/jame.20038, 2013.

Goll, D. S., Brovkin, V., Liski, J., Raddatz, T., Thum, T., and Todd-Brown, K. E. O.: Strong dependence of CO2 emissions from anthropogenic land cover change on initial land cover and soilcarbon parametrization, Global Biogeochem. Cycles, 29, 1511–1523, https://doi.org/10.1002/2014GB004988., 2015.

Gordon, A., Sprintall, J., Van Aken, H., Susanto, D., Wijffels, S., Molcard, R., Ffield, A., Pranowo, W., and Wirasantosa, S.: The Indonesian throughflow during 2004–2006 as observed by the INSTANT program, Dyn. Atmos. Ocean., 50, 115–128, https://doi.org/10.1016/J.DYNATMOCE.2009.12.002, 2010.





Greatbatch, R. J.: The north Atlantic oscillation, Stochastic Environmental Research and Risk Assessment, 14, 213–242, 2000.

Greatbatch, R. J., Zhai, X., Claus, M., Czeschel, L., and Rath, W.: Transport driven by eddy momentum fluxes in the Gulf Stream Extension region, Geophysical Research Letters, 37, L24 401, https://doi.org/10.1029/2010GL045473, 2010.

Griffies, S. et al.: Impacts on Ocean Heat from Transient Mesoscale Eddies in a Hierarchy of Climate Models, J. Climate, 28, 952–976, https://doi.org/10.1175/JCLI-D-14-00-353.1, 2015.

Groner, V. P., Raddatz, T., Reick, C. H., and Claussen, M.: Plant functional diversity affects climate–vegetation interaction, Biogeosciences, 15, 1947–1968, https://doi.org/10.5194/bg-15-1947-2018, 2018.

Haarsma, R. J., Roberts, M. J., Vidale, P. L., Senior, C. A., Bellucci, A., Bao, Q., Chang, P., Corti, S., Fučkar, N. S., Guemas, V., von Hardenberg, J., Hazeleger, W., Kodama, C., Koenigk, T., Leung, L. R., Lu, J., Luo, J.-J., Mao, J., Mizielinski, M. S., Mizuta, R., Nobre, P., Satoh, M., Scoccimarro, E., Semmler, T., Small, J., and von Storch, J.-S.: High Resolution Model Intercomparison Project (HighResMIP v1.0) for CMIP6, Geosci. Model Dev., 9, 4185–4208, https://doi.org/10.5194/gmd-9-4185-2016, 2016.

Haase, S. and Matthes, K.: The importance of interactive chemistry for stratosphere–troposphere coupling, Atmospheric Chemistry and Physics, 19, 3417–3432, https://doi.org/10.5194/acp-19-3417-2019, 2019.

Haase, S., Matthes, K., Latif, M., and Omrani, N. E.: The importance of a properly represented stratosphere for Northern Hemisphere surface variability in the atmosphere and the ocean, Journal of Climate, 31, 8481–8497, https://doi.org/10.1175/JCLI-D-17-0520.1, 2018.

Hagemann, S. and Dümenil Gates, L.: Improving a Subgrid Runoff Parameterization Scheme for Climate Models by the Use of High Resolution Data Derived from Satellite Observations, Climate Dynamics, pp. 349–359, 2003.

Hansen, F., Matthes, K., and Wahl, S.: Tropospheric QBO–ENSO Interactions and Differences between the Atlantic and Pacific, Journal of Climate, 29, 1353–1368, https://doi.org/10.1175/JCLI-D-15-0164.1, 2016.

Harlaß, J., Latif, M., and Park, W.: Improving climate model simulation of tropical Atlantic sea surface temperature: The importance of enhanced vertical atmosphere model resolution, Geophysical Research Letters, 42, 2401–2408, https://doi.org/10.1002/2015GL063310, 2015.

Harlaß, J., Latif, M., and Park, W.: Alleviating tropical Atlantic sector biases in the Kiel climate model by enhancing horizontal and vertical atmosphere model resolution: climatology and interannual variability, Climate Dynamics, 50, 2605–2635, https://doi.org/10.1007/s00382-017-3760-4, 2018.

Hegglin, M. I., Kinnison, D., Plummer, D., and et al.: Historical and future ozone database (1850–2100) in support of CMIP6, Geoscientific Model Development Discussions, in preparation, 2018.

Hibler III, W. D.: A dynamic thermodynamic sea ice model, Journal of Physical Oceanography, 9, 815–846, 1979.

Hines, C. O.: Doppler-spread parameterization of gravity-wave momentum deposition in the middle atmosphere. Part 1: Basic formulation, J. Atmos. Sol. Terr. Phys., 59, 371–386, 1990a.

Hines, C. O.: Doppler-spread parameterization of gravity-wave momentum deposition in the middle atmosphere. Part 2: Broad and quasi monochromatic spectra, and implementation, J. Atmos. Sol. Terr. Phys., 59, 387–400, 1990b.

Hobbs, W., Palmer, M. D., and Monselesan, D.: An Energy Conservation Analysis of Ocean Drift in the CMIP5 Global Coupled Models, Journal of Climate, 29, 1639–1653, https://doi.org/10.1175/JCLI-D-15-0477.1, 2016.

Holton, J. R. and Tan, H.-C.: The Influence of the Equatorial Quasi-Biennial Oscillation on the Global Circulation at 50 mb, Journal of the Atmospheric Sciences, 37, 2200–2208, https://doi.org/10.2151/jmsj1965.60.1_140, 1980.

Holton, J. R. and Tan, H.-C.: The Quasi-Biennial Oscillation in the Northern Hemisphere Lower Stratosphere, Journal of the Meteorological Society of Japan. Ser. II, 60, 140–148, https://doi.org/10.2151/jmsj1965.60.1_140, 1982.





Hurrell, J. W., Kushnir, Y., Ottersen, G., and Visbeck, M.: An overview of the North Atlantic oscillation, Geophysical Monograph 134, American Geophysical Union, Washington, D.C., 2003.

Iacono, M. J., Delamere, J. S., Mlawer, E. J., Shephard, M. W., Clough, S. A., and Collins, W. D.: Radiative forcing by long-lived greenhouse gases: Calculations with the AER radiative transfer models, Journal of Geophysical Research Atmospheres, 113, 2–9, https://doi.org/10.1029/2008JD009944, 2008.

Jones, P.: Conservative remapping: First-and second-order conservative remapping, Mon Weather Rev, 127, 2204–2210, 1999.

Karpechko, A.Y., P. H. D. P. A. S.: Predictability of downward propagation of major sudden stratospheric warmings, Q.J.R.Meteorol.Soc., 143, 1459–1470, https://doi.org/10.1002/qj.3017, 2017.

Khatiwala, S.: A computational framework for simulation of biogeochemical tracers in the ocean, Global Biogeochemical Cycles, 21, https://doi.org/10.1029/2007GB002923, 2007.

Khodri, M., Izumo, T., Vialard, J., Janicot, S., Cassou, C., Lengaigne, M., Mignot, J., Gastineau, G., Guilyardi, E., Lebas, N., Robock, A., and McPhaden, M. J.: Tropical explosive volcanic eruptions can trigger El Ninõ by cooling tropical Africa, Nature Communications, 8, 1–12, https://doi.org/10.1038/s41467-017-00755-6, 2017.

Khon, V. C., Schneider, B., Latif, M., Park, W., and Wengel, C.: CEvolution of Eastern Equatorial Pacific Seasonal and Interannual Variability in Response to Orbital Forcing During the Holocene and Eemian From Model Simulations, Geophysical Research Letters, 45, 9843–9851, https://doi.org/10.1029/2018gl079337, 2018.

Kim, S. T., Cai, W., Jin, F. F., and Yu, J. Y.: ENSO stability in coupled climate models and its association with mean state, Climate Dynamics, 42, 3313–3321, https://doi.org/10.1007/s00382-013-1833-6, 2014.

Krebs, U., a. W. P. and Schneider, B.: Pliocene aridification of Australia caused by tectonically induced weakening of the Indonesian through-flow, Palaeogeography Palaeoclimatology Palaeoecology, 309, 111–117, https://doi.org/10.1016/j.palaeo.2011.06.002, 2011.

Kriest, I.: Calibration of a simple and a complex model of global marine biogeochemistry, Biogeosciences, 14, 4965–4984, https://doi.org/10.5194/bg-14-4965-2017, 2017.

Kriest, I. and Oschlies, A.: MOPS-1.0: Towards a model for the regulation of the global oceanic nitrogen budget by marine biogeochemical processes, Geoscientific Model Development, 8, 2929–2957, https://doi.org/10.5194/gmd-8-2929-2015, 2015.

Kriest, I., Sauerland, V., Khatiwala, S., Srivastav, A., and Oschlies, A.: Calibrating a global three-dimensional biogeochemical ocean model (MOPS-1.0), Geoscientific Model Development, 10, 127–154, https://doi.org/10.5194/gmd-10-127-2017, 2017.

Kwok, R. and Rothrock, D. A.: Decline in Arctic sea ice thickness from submarine and ICESat records: 1958–2008, Geophysical Research Letters, 36, https://doi.org/10.1029/2009GL039035, 2009.

Labitzke, K. and Naujokat, B.: The Lower Arctic Stratosphere in Winter since 1952, 2000.

Latif, M., Semenov, V. A., and Park, W.: Super El Niños in response to global warming in a climate model, Climatic Change, 132, 489–500, 2015.

Loveday, B. R., Durgadoo, J. V., Reason, C. J. C., Biastoch, A., and Penven, P.: Decoupling of the Agulhas Leakage from the Agulhas Current, Journal of Physical Oceanography, 44, 1776–1797, https://doi.org/10.1175/JPO-D-13-093.1, 2014.

Lu, B., Jin, F. F., and Ren, H. L.: A coupled dynamic index for ENSO periodicity, Journal of Climate, 31, 2361–2376, https://doi.org/10.1175/JCLI-D-17-0466.1, 2018.

Lübbecke, J. F., Durgadoo, J. V., and Biastoch, A.: Contribution of Increased Agulhas Leakage to Tropical Atlantic Warming, Journal of Climate, 28, 9697–9706, https://doi.org/10.1175/JCLI-D-15-0258.1, 2015.





Lubis, S. W., Silverman, V., Matthes, K., Harnik, N., Omrani, N. E., and Wahl, S.: How does downward planetary wave coupling affect polar stratospheric ozone in the Arctic winter stratosphere?, Atmospheric Chemistry and Physics, 17, 2437–2458, https://doi.org/10.5194/acp-17-2437-2017, 2017.

Lucarini, V. and Ragone, F.: ENERGETICS OF CLIMATE MODELS: NET ENERGY BALANCE AND MERIDIONAL ENTHALPY TRANSPORT, Reviews of Geophysics, 49, RG1001, https://doi.org/10.1029/2009RG000323, 2011.

Ma, X., Jing, Z., Chang, P., Liu, X., Montuoro, R., Small, R. J., Bryan, F. O., Greatbatch, R. J., Brandt, P., Wu, D., Lin, X., and Wu, L.: Western boundary currents regulated by interaction between ocean eddies and the atmosphere, Nature, 535, 533–-537, https://doi.org/10.1038/nature18640, 2016.

Madec, G.: NEMO ocean engine, Note du Pôle modélisation, Inst. Pierre-Simon Laplace, p. 406, 2016.

Madec, G. and Imbard, M.: A global ocean mesh to overcome the North Pole singularity, Climate Dynamics, 12, 381–388, https://doi.org/10.1007/BF00211684, 1996.

Martin, T., Park, W., and Latif, M.: Multi-centennial variability controlled by Southern Ocean convection in the Kiel Climate Model, Climate Dynamics, 40, 2005–2022, https://doi.org/10.1007/s00382-012-1586-7, 2013.

Martin, T., Park, W., and Latif, M.: Southern Ocean forcing of the North Atlantic at multi-centennial time scales in the Kiel Climate Model, Deep Sea Research II, pp. 39 – 48, https://doi.org/10.1016/j.dsr2.2014.01.018, southern Ocean Dynamics and Biogeochemistry in a Changing Climate, 2015.

Matsuno, T.: A Dynamical Model of the Stratospheric Sudden Warming, Journal of the Atmospheric Sciences, 28, 1479–1494, 1971.

Matthes, K., Funke, B., Andersson, M. E., Barnard, L., Beer, J., Charbonneau, P., Clilverd, M. A., Dudok De Wit, T., Haberreiter, M., Hendry, A., Jackman, C. H., Kretzschmar, M., Kruschke, T., Kunze, M., Langematz, U., Marsh, D. R., Maycock, A. C., Misios, S., Rodger, C. J., Scaife, A. A., Seppälä, A., Shangguan, M., Sinnhuber, M., Tourpali, K., Usoskin, I., Van De Kamp, M., Verronen, P. T., and Versick, S.: Solar forcing for CMIP6 (v3.2), Geoscientific Model Development, 10, 2247–2302, https://doi.org/10.5194/gmd-10-2247-2017, 2017.

Mauritsen, T., B. J. B. T. B. J. B. M. B. R. e. a.: Developments in the MPI-M Earth System Model version 1.2 (MPI-ESM1.2) and its response to increasing CO2, Journal of Advances in Modeling Earth Systems, 11, 998–1038, https://doi.org/10.1029/2018MS001400, 2019.

Mauritsen, T., Stevens, B., Roeckner, E., Crueger, T., Esch, M., Giorgetta, M., Haak, H., Jungclaus, J., Klocke, D., Matei, D., Mikolajewicz, U., Notz, D., Pincus, R., Schmidt, H., and Tomassini, L.: Tuning the climate of a global model, Journal of Advances in Modeling Earth Systems, 4, n/a–n/a, https://doi.org/10.1029/2012MS000154, 2012.

McCarthy, G., Smeed, D., Johns, W., Frajka-Williams, E., Moat, B., Rayner, D., Baringer, M., Meinen, C., Collins, J., and Bryden, H.: Measuring the Atlantic Meridional Overturning Circulation at 26°N, Prog. Oceanogr., 130, 91–111, https://doi.org/10.1016/J.POCEAN.2014.10.006, 2015.

Meijers, A. J. S., Shuckburgh, E., Bruneau, N., Sallee, J.-B., Bracegirdle, T. J., and Wang, Z.: Representation of the Antarctic Circumpolar Current in the CMIP5 climate models and future changes under warming scenarios, Journal of Geophysical Research: Oceans, 117, https://doi.org/10.1029/2012JC008412, 2012.

Meinen, C. S., Baringer, M. O., and Garcia, R. F.: Florida Current transport variability: An analysis of annual and longer-period signals, Deep Sea Res. Part I Oceanogr. Res. Pap., 57, 835–846, https://doi.org/10.1016/J.DSR.2010.04.001, 2010.

Miller, M. J., Palmer, T. N., and Swinbank, R.: Parametrization and influence of subgridscale orography in general circulation and numerical weather prediction models, Meteorology and Atmospheric Physics, 40, 84–109, https://doi.org/10.1007/BF01027469, 1989.

Minobe, S., Kuwano-Yoshida, A., Komori, N., Xie, S.-P., and Small, R. J.: Influence of the Gulf Stream on the troposphere, Nature, 452, 206–-209, https://doi.org/10.1038/nature06690, 2008.





Mitchell, D. M., Gray, L. J., Anstey, J., Baldwin, M. P., Charlton-Perez, A. J., Mitchell, D. M., Gray, L. J., Anstey, J., Baldwin, M. P., and Charlton-Perez, A. J.: The Influence of Stratospheric Vortex Displacements and Splits on Surface Climate, Journal of Climate, 26, 2668–2682, https://doi.org/10.1175/JCLI-D-12-00030.1, 2013.

Mitchell, D. M., Misios, S., Gray, L. J., Tourpali, K., Matthes, K., Hood, L., Schmidt, H., Chiodo, G., Thiéblemont, R., Rozanov, E., Shindell, D., and Krivolutsky, A.: Solar signals in CMIP-5 simulations: The stratospheric pathway, Quarterly Journal of the Royal Meteorological Society, 141, 2390–2403, https://doi.org/10.1002/qj.2530, 2015.

Müller, W. A., Jungclaus, J. H., Mauritsen, T., Baehr, J., Bittner, M., Budich, R., Bunzel, F., Esch, M., Ghosh, R., Haak, H., Ilyina, T., Kleine, T., Kornblueh, L., Li, H., Modali, K., Notz, D., Pohlmann, H., Roeckner, E., Stemmler, I., Tian, F., and Marotzke, J.: A Higher-resolution Version of the Max Planck Institute Earth System Model (MPI-ESM1.2-HR), Journal of Advances in Modeling Earth Systems, 10, 1383–1413, https://doi.org/10.1029/2017MS001217, 2018.

Neelin, J. D., Battisti, D. S., Hirst, A. C., Jin, F.-F., Wakata, Y., Yamagata, T., and Zebiak, S. E.: ENSO theory, Journal of Geophysical Research, 103, 14 261, https://doi.org/10.1029/97JC03424, 1998.

Nordeng, T. E.: Extended versions of the convective parameterization scheme at ECMWF and their impact on the mean and transient activity of the model in the tropics, Tech. Rep. 206, ECMWF, Reading, U. K., 1994.

Omrani, N.-E., Ogawa, F., Nakamura, H., Keenlyside, N., Lubis, S., and Matthes, K.: Key role of the Ocean Western Boundary Currents in shaping the Northern Hemisphere Climate, Nature Scientific Reports, 9, https://doi.org/10.1038/s41598-019-39392-y, 2019.

Orr, J. C. and Epitalon, J.-M.: Improved routines to model the ocean carbonate system: mocsy 2.0, Geoscientific Model Development, 8, 485–499, https://doi.org/10.5194/gmd-8-485-2015, 2015.

Orr, J. C., Najjar, R. G., Aumont, O., Bopp, L., Bullister, J. L., Danabasoglu, G., Doney, S. C., Dunne, J. P., Dutay, J.-C., Graven, H., Griffies, S. M., John, J. G., Joos, F., Levin, I., Lindsay, K., Matear, R. J., McKinley, G. A., Mouchet, A., Oschlies, A., Romanou, A., Schlitzer, R., Tagliabue, A., Tanhua, T., and Yool, A.: Biogeochemical protocols and diagnostics for the CMIP6 Ocean Model Intercomparison Project (OMIP), Geoscientific Model Development, 10, 2169–2199, https://doi.org/10.5194/gmd-10-2169-2017, 2017.

Palmer, M. D. and McNeall, D. J.: Internal variability of Earth's energy budget simulated by CMIP5 climate models, Environmental Research Letters, 9, 034 016, https://doi.org/10.1088/1748-9326/9/3/034016, 2014.

Palmer, T., Shutts, G., and Swinbank, R.: Alleviation of a systematic westerly bias in general circulation and numerical weather prediction models through an orographic gravity wave drag parametrization, Quarterly Journal of the Royal Meteorological Society, 112, 1001–1039, 1986.

Park, W. and Latif, M.: Multidecadal and multicentennial variability of the meridional overturning circulation, Geophysical Research Letters, 35, L22 703, https://doi.org/10.1029/2008GL035779, 2008.

Park, W., Keenlyside, N., Latif, M., Ströh, A., Redler, R., Roeckner, E., Madec, G., Park, W., Keenlyside, N., Latif, M., Ströh, A., Redler, R., Roeckner, E., and Madec, G.: Tropical Pacific Climate and Its Response to Global Warming in the Kiel Climate Model, Journal of Climate, 22, 71–92, https://doi.org/10.1175/2008JCLI2261.1, 2009.

Perlwitz, J.: Tug of war on the jet stream, Nat. Clim. Chang., 1, 29–31, 2011.

Philander, S.: El Niño, La Niña, and the southern oscillation, vol. 46, Academic Press, San Diego, USA, 1990.

Quiroz, R. S.: The tropospheric-stratospheric polar vortex breakdown of January 1977, Geophysical Research Letters, 4, 151–154, https://doi.org/10.1029/GL004i004p00151, 1977.





Rayner, N. A., Parker, D. E., Horton, E. B., Folland, C. K., Alexander, L. V., Rowell, D. P., Kent, E. C., and Kaplan, A.: Global analyses of sea surface temperature, sea ice, and night marine air temperature since the late nineteenth century, Journal of Geophysical Research: Atmospheres, 108, 4407, https://doi.org/10.1029/2002JD002670, 2003.

Reichler, T., Kim, J., Manzini, E., and Kröger, J.: A stratospheric connection to Atlantic climate variability, Nature Geoscience, 5, 783–787, https://doi.org/10.1038/ngeo1586, 2012.

Reick, C. H., Raddatz, T., Brovkin, V., and Gayler, V.: epresentation of natural and anthropogenic land cover change in MPI-ESM, Adv. Model. Earth Sy., 5, 459–482, https://doi.org/10.1002/jame.20022, 2013.

Renault, L. et al.: Control and Stabilization of the Gulf Stream by Oceanic Current Interaction with the Atmosphere, J. Phys. Ocean., 46, 3439–3453, https://doi.org/10.1175/JPO-D-16-0115.1, 2016.

Richardson, P. L.: Agulhas leakage into the Atlantic estimated with subsurface floats and surface drifters, Deep. Res. I, 54, 1361–1389, 2007.

Richter, I.: Climate model biases in the eastern tropical oceans: causes, impacts and ways forward, Wiley Interdisciplinary Reviews: Climate Change, 6, 345–358, https://doi.org/10.1002/wcc.338, 2015.

Richter, I. and Xie, S.-P.: On the origin of equatorial Atlantic biases in coupled general circulation models, Climate Dynamics, 31, 587–598, https://doi.org/10.1007/s00382-008-0364-z, 2008.

Richter, I., Xie, S.-P., Behera, S. K., Doi, T., and Masumoto, Y.: Equatorial Atlantic variability and its relation to mean state biases in CMIP5, Climate Dynamics, 42, 171–188, https://doi.org/10.1007/s00382-012-1624-5, 2012.

Ridgway, K. and Dunn, J.: Observational evidence for a Southern Hemisphere oceanic supergyre., Geophys. Res. Lett., 34, L1–5, https://doi.org/10.1029/2007GL030392, 2007.

Roemmich, D., Church, J., Gilson, J., Monselesan, D., Sutton, P., and Wijffels, S.: Unabated planetary warming and its ocean structure since 2006, Nature climate change, 5, 240, https://doi.org/10.1038/nclimate2513, 2015.

Scherhag, R.: Die explosionsartigen Stratospherenerwarmingen des Spatwinters 1951-1952, 1952.

Schmidt, H., Brasseur, G., Charron, M., Manzini, E., Giorgetta, M., Diehl, T., Fomichev, V. I., Kinnison, D., Marsh, D., and Walters, S.: The HAMMONIA Chemistry Climate Model: Sensitivity of the Mesopause Region to the 11-Year Solar Cycle and CO 2 Doubling, Journal of Climate, 19, 3903 – 3931, 2006.

Schmidt, H., Rast, S., Bunzel, F., Esch, M., Giorgetta, M., Kinne, S., Krismer, T., Stenchikov, G., Timmreck, C., Tomassini, L., and Walz, M.: Response of the middle atmosphere to anthropogenic and natural forcings in the CMIP5 simulations with the Max Planck Institute Earth system model, Journal of Advances in Modeling Earth Systems, 5, 98–116, https://doi.org/10.1002/jame.20014, 2013.

Schmittner, A., Oschlies, A., Matthews, H. D., and Galbraith, E.: Future changes in climate, ocean circulation,ecosystems, and biogeochemical cycling simulated for a business-as-usual CO2 emission scenario until year 4000 AD., Glob. Biogeochem. Cycles, 22, GB1013, https://doi.org/doi:10.1029/2007GB002953, 2008.

Schneider, B., Leduc, G., and Park, W.: Disentangling seasonal signals in Holocene climate trends by satellite-model-proxy integration, Paleoceanography, 25, https://doi.org/10.1029/2009pa001893, 2010.

Schultz, M. G., Stadtler, S., Schröder, S., Taraborrelli, D., Franco, B., Krefting, J., Henrot, A., Ferrachat, S., Lohmann, U., Neubauer, D., Siegenthaler-Le Drian, C., Wahl, S., Kokkola, H., Kühn, T., Rast, S., Schmidt, H., Stier, P., Kinnison, D., Tyndall, G. S., Orlando, J. J., and Wespes, C.: The chemistry-climate model ECHAM6.3-HAM2.3-MOZ1.0, Geoscientific Model Development, 11, 1695–1723, https://doi.org/10.5194/gmd-11-1695-2018, 2018.





Schwarzkopf, F. U., Biastoch, A., Böning, C. W., Chanut, J., Durgadoo, J. V., Getzlaff, K., Harlaß, J., Rieck, J. K., Roth, C., Scheinert, M. M., and Schubert, R.: The INALT family – a set of high-resolution nests for the Agulhas Current system within global NEMO ocean/sea-ice configurations, Geoscientific Model Development, 12, 3329–3355, https://doi.org/10.5194/gmd-12-3329-2019, 2019.

Schweiger, A., Lindsay, R., Zhang, J., Steele, M., Stern, H., and Kwok, R.: Uncertainty in modeled Arctic sea ice volume, Journal of Geophysical Research: Oceans, 116, https://doi.org/10.1029/2011JC007084, 2011.

Sen Gupta, A., Santoso, A., Taschetto, A. S., Ummenhofer, C. C., Trevena, J., and England, M. H.: Projected Changes to the Southern Hemisphere Ocean and Sea Ice in the IPCC AR4 Climate Models, Journal of Climate, 22, 3047–3078, https://doi.org/10.1175/2008JCLI2827.1, 2009.

Sen Gupta, A., Muir, L., Brown, J., Phipps, S., Durack, P., Monselesan, D., and S, W.: Climate Drift in CMIP3 Models, Journal of Climate, 25, 4621–4640, https://doi.org/10.1175/JCLI-D-11-00312.1, 2012.

Shaw, T. A., Perlwitz, J., and Weiner, O.: Troposphere-stratosphere coupling: Links to North Atlantic weather and climate, including their representation in CMIP5 models, Journal of Geophysical Research, 119, 5864–5880, https://doi.org/10.1002/2013JD021191, 2014.

Shu, Q., Song, Z., and Qiao, F.: Assessment of sea ice simulations in the CMIP5 models, The Cryosphere, 9, 399–409, https://doi.org/10.5194/tc-9-399-2015, 2015.

Small, R. J. and et al.: A new synoptic scale resolving global climate simulation using the Community Earth System Model, J. Adv. Model. Earth Syst., 6, 1065—-1094, https://doi.org/10.1002/2014MS000363, 2014.

Small, R. J., Curchitser, E., Hedstrom, K., Kauffman, B., and Large, W. G.: The Benguela Upwelling System: Quantifying the Sensitivity to Resolution and Coastal Wind Representation in a Global Climate Model*, Journal of Climate, 28, 9409–9432, https://doi.org/10.1175/JCLI-D-15-0192.1, 2015.

Song, Z., Latif, M., and Park, W.: East Atlantic Pattern Drives Multidecadal Atlantic Meridional Overturning Circulation Variability during the Last Glacial Maximum, Geophysical Research Letters, in revision, https://doi.org/?, 2019.

Song, Z. Y., Latif, M., Park, W., Krebs-Kanzow, U., and Schneider, B.: Influence of seaway changes during the Pliocene on tropical Pacific climate in the Kiel climate model: mean state, annual cycle, ENSO, and their interactions, Climate Dynamics, 48, 3725–3740, https://doi.org/10.1007/s00382-016-3298-x, 2017.

Steele, M., Morley, R., and Ermold, W.: PHC: A Global Ocean Hygrography with a High-Quality Arctic Ocean., J. Clim., 14, 2079–2087, 2001.

Stenchikov, G. L., Kirchner, I., Robock, A., Graf, H.-F., Antuña, J. C., Grainger, R. G., Lambert, A., and Thomason, L.: Radiative forcing from the 1991 Mount Pinatubo volcanic eruption, Journal of Geophysical Research, 103, 13 837–13 857, https://doi.org/10.1029/98JD00693, 1998.

Stevens, B., Giorgetta, M., Esch, M., Mauritsen, T., Crueger, T., Rast, S., Salzmann, M., Schmidt, H., Bader, J., Block, K., Brokopf, R., Fast, I., Kinne, S., Kornblueh, L., Lohmann, U., Pincus, R., Reichler, T., and Roeckner, E.: Atmospheric component of the MPI-M Earth System Model: ECHAM6, Journal of Advances in Modeling Earth Systems, 5, 146–172, https://doi.org/10.1002/jame.20015, 2013.

Stevens, B., Fiedler, S., Kinne, S., Peters, K., Rast, S., M?sse, J., Smith, S. J., and Mauritsen, T.: MACv2-SP: a parameterization of anthropogenic aerosol optical properties and an associated Twomey effect for use in CMIP6, Geoscientific Model Development, 10, 433–452, https://doi.org/10.5194/gmd-10-433-2017, 2017.

Stroeve, J. C., Kattsov, V., Barrett, A., Serreze, M., Pavlova, T., Holland, M., and Meier, W. N.: Trends in Arctic sea ice extent from CMIP5, CMIP3 and observations, Geophysical Research Letters, 39, L16 502, https://doi.org/10.1029/2012GL052676, 2012.





Thiéblemont, R., Matthes, K., Omrani, N.-E., Kodera, K., and Hansen, F.: Solar forcing synchronizes decadal North Atlantic climate variability, Nat. Commun., 6, https://doi.org/10.1038/ncomms9268, 2015.

Thompson, D. W., Baldwin, M. P., and Wallace, J. M.: Stratospheric connection to Northern Hemisphere wintertime weather: Implications for prediction, Journal of Climate, 15, 1421–1428, https://doi.org/10.1175/1520-0442(2002)015<1421:SCTNHW>2.0.CO;2, 2002.

Tiedtke, M.: A Comprehensive Mass Flux Scheme for Cumulus Parameterization in Large-Scale Models, Monthly Weather Review, 117, 1779–1800, https://doi.org/10.1175/1520-0493(1989)117<1779:ACMFSF>2.0.CO;2, 1989.

Timmermann, A., An, S.-i., Kug, J.-s., Jin, F.-f., Cai, W., Cobb, K., Lengaigne, M., Mcphaden, M. J., Malte, F., Stein, K., Wittenberg, A. T., Yun, K.-s., Bayr, T., Chen, H.-c., Chikamoto, Y., Dewitte, B., Dommenget, D., Grothe, P., Ham, Y.-g., Hayashi, M., Ineson, S., Kang, D., Kim, W., Lee, J.-y., Li, T., Luo, J.-j., Mcgregor, S., Power, S., Rashid, H., Ren, H.-l., Santoso, A., Takahashi, K., Todd, A., Wang, G.,
Wang, G., Xie, R., Yang, W.-h., Yeh, W., Yoon, J., Zeller, E., and Zhang, X.: El Niño-Southern Oscillation Complexity, Nature, pp. 1–25, https://doi.org/10.1038/s41586-018-0252-6, 2018.

Tozuka, T., Doi, T., Miyasaka, T., Keenlyside, N., and Yamagata, T.: Key factors in simulating the equatorial Atlantic zonal sea surface temperature gradient in a coupled general circulation model, Journal of Geophysical Research, 116, 1–12, https://doi.org/10.1029/2010JC006717, 2011.

Treguier, A. M., Held, I. M., Larichev, V. D., Treguier, A. M., Held, I. M., and Larichev, V. D.: Parameterization of Quasigeostrophic Eddies in Primitive Equation Ocean Models, J. Phys. Oceanogr., 27, 567–580, https://doi.org/10.1175/1520-0485, 1997.

Ullgren, J., van Aken, H., Ridderinkhof, H., and de Ruijter, W.: The hydrography of the Mozambique Channel from six years of continuous temperature, salinity, and velocity observations, Deep Sea Res. Part I Oceanogr. Res. Pap., 69, 36–50, https://doi.org/10.1016/j.dsr.2012.07.003, 2012.

Valcke, S.: The OASIS3 coupler: a European climate modelling community software, Geoscientific Model Development, 6, 373–388, https://doi.org/10.5194/gmd-6-373-2013, 2013.

Voldoire, A., Claudon, M., Caniaux, G., Giordani, H., and Roehrig, R.: Are atmospheric biases responsible for the tropical Atlantic SST biases in the CNRM-CM5 coupled model?, Climate Dynamics, 43, 2963–2984, https://doi.org/10.1007/s00382-013-2036-x, 2014.

Wahl, S., Latif, M., Park, W., and Keenlyside, N.: On the Tropical Atlantic SST warm bias in the Kiel Climate Model, Climate Dynamics,
36, 891–906, https://doi.org/10.1007/s00382-009-0690-9, 2009.

Wang, C., Zhang, L., Lee, S.-K., Wu, L., and Mechoso, C. R.: A global perspective on CMIP5 climate model biases, Nature Climate Change, 4, 201–205, https://doi.org/10.1038/nclimate2118, 2014.

Watterson, I. G., Bathols, J., and Heady, C.: What influences the skill of climate models over the continents?, B. Am. Meteorol. Soc., 95, 689—-700, https://doi.org/10.1175/BAMS-D-12-00136.1, 2014.

Welch, P.: The use of the fast Fourier transform for the estimation of power spectra: A method based on time averaging over short, modified periodograms, IEEE Trans. Audio Electroacoust, 15, 70–73, 1967.

Wengel, C., Latif, M., Park, W., Harlaß, J., and Bayr, T.: Seasonal ENSO phase locking in the Kiel Climate Model: The importance of the equatorial cold sea surface temperature bias, Climate Dynamics, 50, 901–919, https://doi.org/10.1007/s00382-017-3648-3, 2018.

Williams, K. D. and et al.: The Met Office Global Coupled model 3.0 and 3.1 (GC3.0 and GC3.1) configurations, J. Adv. Model. Earth Syst.,
10, 357—-380, https://doi.org/10.1002/2017MS001115, 2017.

Xu, Z., Chang, P., Richter, I., Kim, W., and Tang, G.: Diagnosing southeast tropical Atlantic SST and ocean circulation biases in the CMIP5 ensemble, Climate Dynamics, 43, 3123–3145, https://doi.org/10.1007/s00382-014-2247-9, 2014.

Zalesak, S. T.: Fully multidimensional flux corrected transport algorithms for fluids, J. Comput. Phys., 31, 1979.



Zantopp, R., Fischer, J., Visbeck, M., and Karstensen, J.: From interannual to decadal: 17 years of boundary current transports at the exit of the Labrador Sea, J. Geophys. Res. Ocean., https://doi.org/10.1002/2016JC012271, 2017.