# Peer review of "The Flexible Ocean and Climate Infrastructure Version 1 (FOCI1): Mean State and Variability"

_Geoscientific Model Development, 2019_

## Referee Comment (RC1) · Anonymous Referee #1 · 19 Feb 2020

The authors document a new earth-system modelling framework developed from the Kiel Climate Model. A novel capability of the new system is the ability to refine the ocean model resolution in specific regions of interest in the context of the full coupled model. The authors present a quite comprehensive baseline assessment of the new system run in a variety of configurations including some assessment of the impact of the local grid refinement in the ocean.

I thought the paper was well written and structured, the figures are clear and the results well presented. I am therefore recommending that it is accepted subject to a few minor revisions, mainly to do with clarity.

p.3 l.16 : Should define/reference MPI-ESM the first time it is mentioned.

p.10 l.5 : "eventually" is an odd word here. Maybe you mean "lastly"?

p.10 l.5 : "This choice depends on the validation of the simulated boundary currents". I'm interested in the criteria used. Is it possible to expand on this in 2 or 3 sentences? (Or reference other work?).

p.12 l.8 : "...as well as the dip in global mean temperature during the first part of the 1990s is captured very well by the ensemble means and might be related to the Pinatubo eruption." A bit garbled - better to split the sentence: "...as well as the dip in global mean temperature during the first part of the 1990s is captured very well by the ensemble means. The dip in the early 1990s is related to the eruption of Mount Pinatubo."

p.23 l.3 : "Under pre-industrial forcing, the AMOC shall not show any long-term trend" : I think you mean "the AMOC *should* not show any long-term trend".

p.23 l.9 : I found it difficult to understand the sentence beginning "While the southward flow...". Maybe something like this would be better: "While the southward return flow reaches down to the depth of classical upper NADW, the depth of classical lower NADW is not connected to the southward flow any more. This is a typical deficit seen in coarse-resolution OGCMs..."

p.27 l.31 : "inevitable" is a strange word in this context. Maybe you mean "required" or "necessary"?

p. 36 l. 7 : "While FOCI produces a realistic simulation of NH sea-ice distribution and evolution, sea ice on the SH is significantly underestimated. The latter bias is subject to ongoing work and likely related to mixing in the upper ocean." This doesn't tie in with what you said in Section 3.3 where you talk about the warm SST bias and the open-ocean deep convection in the Weddell Sea. If you think upper-ocean mixing is also important in the story of the Southern Ocean sea ice biases you should mention this in Section 3.3 rather than just introducing it in the Conclusions.

p. 38 l. 5 : "Therefor" should be "Therefore".

Fig 3 : The caption needs some work. In particular the description of panels (c) and (d) is wrong/incomplete. Both panels include FOCI-hist and FOCI-chem and should state that (c) is SAT and (d) is 10hPa temperature.

Fig 18 : Still have a question mark in the caption.

Fig A5 : Should probably expand "SW" as shortwave and "LW" as longwave in the caption.

[Figure]

---

## Referee Comment (RC2) · Anonymous Referee #2 · 5 Mar 2020

The authors present the documentation and first evaluation of a new Earth System Model, FOCI1. In the model complexity/ model resolution plane for Earth System models (ESMs), FOCI1 occupies a less well populated region due to its comparatively high resolution in the ocean. Hence FOCI1 is a very relevant addition to the field of Earth System modelling.

The paper is written very clearly. It is commendable that the authors are very upfront about the strengths and weaknesses of FOCI1, these are described in detail. The presented evaluation is clear and comprehensive. The gains from a nested high-resolution ocean are clearly visible. It is remarkable that in spite of the relatively coarse resolution in the atmosphere a realistic QBO is simulated.

I'm tempted to accept the ms. as is, but I'll suggest a small number of items to improve

on instead ("minor revisions").

**Detailed remarks**

Introduction, l.10 "... some processes are neglected, such as the upper atmosphere, atmospheric chemistry, ...": This would have been true for the CMIP5 ESMs. However, in the current CMIP6 generation there are very comprehensive ESMs that have a high-top atmosphere and/ or a full stratosphere-troposphere atmospheric chemistry. Examples are CNRM-ESM2-1 and UKESM1. It would be only fair to mention this here. FOCI1 still stands out against the CMIP6 ESMs in having an eddy-rich ocean (when using nesting). In the same vein, some of the current CMIP6 ESMs have fully dynamic aerosols. This is a feature that the HighResMIP models and FOCI1 deliberately do not have. p.11, footnote: I'm curious what "technical reasons" kept you from integrating FOCI1 right through 31-12-2014.

p.13 l.9 "energy is not fully conserved": some headway has been made here in the CMIP6 ESMs by reducing known energy leaks and/ or by enforcing closure. An example is Walters et al. (https://doi.org/10.5194/gmd-12-1909-2019) who discuss the improvements in energy conservation in the UK Met Office atmosphere model. Where does ECHAM6.3 stand in that regard?

p.13, l.10 "imbalance of 0.7 W/m$^2$": this figure is large for a pre-industrial control run, but acceptable for the end of the historical period. Can you give this figure from the FOCI1 simulations for both instances? Or even consider an additional panel in Fig.3 to plot the TOA net imbalance time-series.

p.17, l.14 "we do not expect ... FOCI ... to capture the observed interannual variability": why not?

p. 43: the references are for the whole paper, as opposed to what the headline says.

———————————————

---

## Author Comment (AC1) · 30 Mar 2020

**Reply to comments of Reviewer 1**

*p.3 l.16 : Should define/reference MPI-ESM the first time it is mentioned.*

The text has been adjusted as suggested.

*p.10 l.5 : "eventually" is an odd word here. Maybe you mean "lastly"?*

Yes. We changed the word.

*p.10 l.5 : "This choice depends on the validation of the simulated boundary currents". I'm interested in the criteria used. Is it possible to expand on this in 2 or 3 sentences? (Or reference other work?).*

This paragraph has been expanded to elucidate the chosen boundary conditions: "The latter are changed to no-slip conditions in case of FOCI_INALT10X, but kept at free slip for FOCI_VIKING10. These particular choices depend on the validation of regionally relevant simulated boundary currents. ORCA05 is optimized for the free-slip condition to represent transports through comparatively narrow straits and channels. At higher horizontal resolution, the no-slip condition is regarded as more realistic. In case of FOCI_INALT10X, the vertical structure, strength and variability of two important western boundary currents, namely Agulhas and Malvinas currents, perform better with a no-slip condition in the nested area (Schwarzkopf et al., 2019). In contrast, free-slip conditions yield a more realistic Gulf Stream separation in FOCI_VIKING10 and a better representation of boundary currents in the northern part of the nest region where the resolution is not sufficient to resolve the Rossby radius. However, no-slip conditions are applied to a stretch of the Greenland coast near Cape Desolation in FOCI_VIKING10 to enhance eddy shedding from the West Greenland Current (Rieck et al., 2019)."

*p.12 l.8 : "...as well as the dip in global mean temperature during the first part of the 1990s is captured very well by the ensemble means and might be related to the Pinatubo eruption." A bit garbled - better to split the sentence: "...as well as the dip in global mean temperature during the first part of the 1990s is captured very well by the ensemble means. The dip in the early 1990s is related to the eruption of Mount Pinatubo."*

The sentence has been adjusted according to the suggestion.

*p.23 l.3 : "Under pre-industrial forcing, the AMOC shall not show any long-term trend" : I think you mean "the AMOC *should* not show any long-term trend".*

Yes. We changed the word.

*p.23 l.9 : I found it difficult to understand the sentence beginning "While the southward flow...". Maybe something like this would be better: "While the southward return flow reaches down to the depth of classical upper NADW, the depth of classical lower NADW is not connected to the southward flow any more. This is a typical deficit seen in coarse- resolution OGCMs..."*

We rephrased the sentence accordingly.

*p.27 l.31 : "inevitable" is a strange word in this context. Maybe you mean "required" or "necessary"?*

We opted for the word "essential".

*p. 36 l. 7 : "While FOCI produces a realistic simulation of NH sea-ice distribution and evolution, sea ice on the SH is significantly underestimated. The latter bias is subject to ongoing work and likely related to mixing in the upper ocean." This doesn't tie in with what you said in Section 3.3 where you talk about the warm SST bias and the open-ocean deep convection in the Weddell Sea. If you think upper-ocean mixing is also important in the story of the Southern Ocean sea ice biases you should mention this in Section 3.3 rather than just introducing it in the Conclusions.*

We added our thoughts on upper ocean mixing to Section 3.3 and rephrased the sentence in the Summary.
Section 3.3: "The underestimation of sea-ice thickness and extent are likely related to a pronounced warm bias in the Southern Ocean (Fig. 13c), **partly related to upper ocean mixing,** and the regular recurrence of open ocean deep convection in the Weddell Sea, which is simulated by FOCI on a multi-decadal time scale (not shown)."
Summary (p37, l23-24): "The latter bias is subject to ongoing work and likely related to mixing in the upper ocean tied to Southern Ocean warm bias, likely induced by upper ocean mixing and open ocean deep convection."

*p. 38 l. 5 : "Therefor" should be "Therefore".*

Corrected.

*Fig 3 : The caption needs some work. In particular the description of panels (c) and (d) is wrong/incomplete. Both panels include FOCI-hist and FOCI-chem and should state that (c) is SAT and (d) is 10hPa temperature.*

The caption has been rewritten as "(a) Global surface air temperature and (b) global 10 hPa temperature from FOCI-piCtl. The black bar in (a) highlights year 1000 to 1500 which are used to evaluate the mean state of FOCI throughout this publication. (c) Global surface air temperature and (d) 10 hPa temperature from the FOCI-hist and FOCI-chem experiments from 1950 through 2013. For the historical period in (c,d) the ERA-interim reanalysis temperature is included as an observational estimate."

*Fig 18 : Still have a question mark in the caption.*

Removed. We verified that the QBO was calculated for 50hPa.

*Fig A5 : Should probably expand "SW" as shortwave and "LW" as longwave in the caption.*

Figure caption has been adjusted accordingly.

**Reply to comments of Reviewer 2**

*Introduction, l.10 ". . . some processes are neglected, such as the upper atmosphere, atmospheric chemistry, . . .": This would have been true for the CMIP5 ESMs. However, in the current CMIP6 generation there are very comprehensive ESMs that have a high-top atmosphere and/ or a full stratosphere-troposphere atmospheric chemistry. Examples are CNRM-ESM2-1 and UKESM1. It would be only fair to mention this here. FOCI1 still stands out against the CMIP6 ESMs in having an eddy-rich ocean (when using nesting). In the same vein, some of the current CMIP6 ESMs have fully dynamic aerosols. This is a feature that the HighResMIP models and FOCI1 deliberately do not have.*

The sentence "Due to computational… ...regional scales." has been rephrased and extended to: "Due to computational limits some processes such as covering the full stratosphere and mesosphere including interactive chemistry (e.g., CESM1-WACCM, Marsh et al., 2013) or interactive aerosol (e.g., UKESM1, Sellar et al., 2019) are only included in some of the current CMIP6 models. In the ocean, mesoscale eddies that provide an important contribution not only to the ocean circulation but also to atmosphere-ocean interactions on global and particularly regional scales are not

resolved in any of the CMIP6 models except for some model configurations used in HighResMIP (Haarsma et al., 2016)."

*p.11, footnote: I'm curious what "technical reasons" kept you from integrating FOCI1 right through 31-12-2014.*

ECHAM6 needs to know about e.g. ozone concentrations for the following year to be able to perform online interpolations. At the time of running the simulations e.g. ozone data for the scenarios starting in 2015 was not available, and hence the model couldn't simulate year 2014. This issue is solved by now.

*p.13 l.9 "energy is not fully conserved": some headway has been made here in the CMIP6 ESMs by reducing known energy leaks and/ or by enforcing closure. An example is Walters et al. (https://doi.org/10.5194/gmd-12-1909-2019) who discuss the improvements in energy conservation in the UK Met Office atmosphere model. Where does ECHAM6.3 stand in that regard?*

A lot of effort has been put into ECHAM6 to investigate energy leakage in ECHAM6.3 which is summarized in Mauritsen et al., 2012, page 8: "We investigated the leakage of energy in MPI-ESM-LR of about 0.5 $W/m^2$ and found that it arises for the most part from mismatching grids and coastlines between the atmosphere and ocean model components. Further, some energy is lost due to an inconsistent treatment of the temperature of precipitation and river runoff into the ocean, and a small leakage of about 0.05 $Wm^2$ occurs in a not yet identified part of the atmosphere…". While developing FOCI we have put a lot of effort into reducing the land-sea mask mismatch between ECHAM6.3 and NEMO3.6 especially given the large difference in atmosphere and ocean resolution in FOCI in it's default setup.

*p.13, l.10 "imbalance of 0.7 W/m2": this figure is large for a pre-industrial control run, but acceptable for the end of the historical period. Can you give this figure from the FOCI1 simulations for both instances? Or even consider an additional panel in Fig.3 to plot the TOA net imbalance time-series.*

We added an extra panel in Figure 3 (c,f) and rephrased p13, l10 to discuss the figure: "The current version of FOCI shows a radiation imbalance of +0.66 $W/m^2$ which is in the range of values we see in other climate models (Mauritsen et al. (2012); CMIP3, Lucarini and Ragone (2011); CMIP5, Hobbs et al. (2016)) but larger than the values achieved in the latest version of MPI-ESM (Müller et al., 2018) which uses the same atmosphere model (but at a higher horizontal resolution). Considering that the evolution of the SAT is relatively stable (Figure 3c) in FOCI-piCtl, we expect a gradual warming of the ocean (see section 3.2). If we subtract the mean TOA radiation imbalance of +0.66 $W/m^2$ derived from

FOCI-piCtl, we find a TOA radiation imbalance of +0.59 W/m$^2$ for the period 1985 to 2012 (Figure 3f) which is in good agreement with the 0.47 Wm$^2$ derived by Allan et al. (2014) from satellite observations for the same period."

*p.17, l.14 "we do not expect . . . FOCI . . . to capture the observed interannual variability": why not?*

To clarify: we expect to capture interannual variability as a statistical measure, but not for individual years/events directly. We rephrased the sentence: "However, we do not expect the free-running FOCI model system to capture the observed interannual variability in terms of particular events, such as the dynamically very specific situation in 2002 and the resulting very weak ozone hole, but rather in a statistical sense, i.e. the distribution of observed variability."

*p. 43: the references are for the whole paper, as opposed to what the headline says.*

Thank you for the hint. We will make sure that the final printed version has the references separated from the appendix.

**New References:**

Haarsma, R. J., Roberts, M. J., Vidale, P. L., Catherine, A., Bellucci, A., Bao, Q., … Von Storch, J. S. (2016). High Resolution Model Intercomparison Project (HighResMIP v1.0) for CMIP6. *Geoscientific Model Development*, *9*(11), 4185–4208. https://doi.org/10.5194/gmd-9-4185-2016

Marsh, D. R., Mills, M. J., Kinnison, D. E., Lamarque, J. F., Calvo, N., & Polvani, L. M. (2013). Climate change from 1850 to 2005 simulated in CESM1(WACCM). *Journal of Climate*, *26*(19), 7372–7391. https://doi.org/10.1175/JCLI-D-12-00558.1

Sellar, A. A., Jones, C. G., Mulcahy, J. P., Tang, Y., Yool, A., Wiltshire, A., … Zerroukat, M. (2019). UKESM1: Description and Evaluation of the U.K. Earth System Model. *Journal of Advances in Modeling Earth Systems*, *11*(12), 4513–4558. https://doi.org/10.1029/2019MS001739